# Bellman Unbiasedness: Toward Provably Efficient Distributional Reinforcement Learning with General Value Function Approximation

## Abstract

Distributional reinforcement learning improves performance by effectively capturing environmental stochasticity. However, existing research on its regret analysis has relied heavily on structural assumptions that are difficult to implement in practice. In particular, there has been little attention to the infeasibility issue of dealing with the infinite-dimensionality of a distribution. To overcome this infeasibility, we present a regret analysis of distributional reinforcement learning with general value function approximation in a finite episodic Markov decision process scenario through *statistical functional dynamic programming*. We first introduce a key notion of *Bellman unbiasedness* which is essential for exactly learnable and provably efficient online updates. Among all types of statistical functionals for representing infinite-dimensional return distributions, our theoretical results demonstrate that only moment functionals can exactly capture the statistical information. Secondly, we propose a provably efficient algorithm, SF-LSVI, that achieves a tight regret bound of $\tilde{O}(d_E H^{\frac{3}{2}} \sqrt{K})$ where $H$ is the horizon, $K$ is the number of episodes, and $d_E$ is the eluder dimension of a function class.

## 1 Introduction

Distributional reinforcement learning (DistRL) (Bellemare et al., 2017; Rowland et al., 2019; Choi et al., 2019; Kim et al., 2024b; Peng et al., 2024; Wiltzer et al., 2024) is an advanced approach to reinforcement learning (RL) that focuses on the entire probability distribution of returns rather than solely on the expected return. By considering the full distribution of returns, distRL provides deeper insight into the uncertainty of each action, such as the mode or median. This framework enables us to make safer and more effective decisions that account for various risks (Chow et al., 2015; Son et al., 2021; Greenberg et al., 2022; Kim et al., 2024a), particularly in complex real-world situations, such as robotic manipulation (Bodnar et al., 2019), neural response (Muller et al., 2024), stratospheric balloon navigation (Bellemare et al., 2020), algorithm discovery (Fawzi et al., 2022), and several game benchmarks (Bellemare et al., 2013; Machado et al., 2018).

In practice, a probability distribution contains an infinite amount of information, and we must inevitably resort to approximations using a finite number of parameters or statistical functionals, such as categorical (Bellemare et al., 2017) and quantile representations (Dabney et al., 2018b). However, previous works have often conducted analyses while overlooking the infeasibility of dealing with the infinite-dimensionality of a distribution. Additionally, not all statistical functionals can be *exactly learned* through the Bellman operator, as the meaning of statistical functionals is not typically preserved after updates. For example, the median is not preserved under Bellman updates, as the median of a mixture of two distributions does not generally correspond to the mixture of their medians. Hence, a distinct analysis is needed to determine whether there exists a corresponding Bellman operator for each statistical functional that guarantees *exactness*.

To address this issue, Rowland et al. (2019) introduced the concept of *Bellman closedness* – a property of statistical functionals that can be exactly learned by the presence of a corresponding Bellman operator. At this point, we take a closer look at the distributional Bellman update. We reconsider what additional properties of statistical functionals beyond Bellman closedness are necessary for constructing online RL algorithms that are not only exactly learnable but also *provably efficient* in terms of regret. Simply, we identify two critical issues when using statistical functionals in lieu of a distribution, which motivated us to address them:

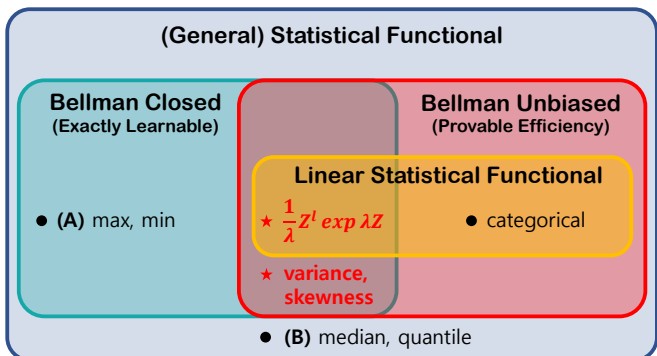

Figure 1: **Venn-Diagram of Statistical Functional Classes.** The diagram illustrates categories of statistical functional. (**Yellow ∩ Blue**) Within the linear statistical functional class, Rowland et al. (2019) demonstrated that the solution of Bellman closedness is uniquely represented by moment functionals. (**Red**) We extend this concept by introducing Bellman unbiasedness, which not only covers moment functionals but also includes central moment functionals from the broader class including nonlinear statistical functionals. (**A**) Maximum and minimum functionals are Bellman closed, while they are not unbiasedly estimatable. (**B**) Median and quantile functionals are neither Bellman closed nor unbiased, highlighting that they are not proper to encode the distribution in terms of exactness. For detailed proof, see Theorem 4.6, Appendix C.2 and C.3.

- **Approximation Error:** Representing a mixture distribution with a finite, fixed number of parameters inevitably leads to approximation errors. For example, when expressing the mixture of two distributions, each represented by $N$ parameters, reducing $2N$ parameters back into $N$ results in inevitable loss of information.

- **Unbiased Estimation:** Due to the unknown nature of the transition $\mathbb{P}(\cdot|s, a)$, the target distribution is estimated through sampling the next state $s'$. Consequently, the statistical functionals of the target distribution (population) should be unbiasedly estimated using the statistical functionals derived from the sample distributions (subpopulations).

In this paper, we introduce a key concept, *Bellman unbiasedness*, for precise information learnability of a distribution from a finite number of samples in an online setting. As illustrated in Figure 1, we prove that the moment functional is the only solution within a class that includes *nonlinear* statistical functionals, satisfying both Bellman closedness and unbiasedness properties. We further discuss the infeasibility of the previously defined structural assumption known as *Distributional Bellman Completeness* (DistBC) (Wang et al., 2023; Chen et al., 2024). We then investigate the benefits of redesigning this concept using a collection of statistical functionals. Finally, we propose a provably efficient statistical functional reinforcement learning algorithm with general value function approximation, called `SF-LSVI`.

In summary, our main contributions are as follows.

- Introduce a key property of Bellman unbiasedness, which is crucial for developing exactly learnable and provably efficient online distRL algorithm, and prove that the moment functional is the unique structure within a class that includes nonlinear statistical functionals.

- Introduce a new fundamental problem related to the inherent infeasibility of dealing with infinite-dimensionalilty of a distribution and provide a theoretically analyzsis of how hidden approximation errors prevent the design of provably efficient algorithms. To address this, we revisit the existing structural assumption of distributional Bellman Completeness through a statistical functional lens.

- Propose a novel, provably efficient distributional RL algorithm called `SF-LSVI`, achieving a tight regret upper bound $\tilde{O}(d_E H^{\frac{3}{2}} \sqrt{K})$. [1] Our framework yields a tighter regret bound while relying on a weaker structural assumption compared to previous results in distributional RL.

---

[1] We ignore poly-log terms in $H, S, A, K$ in the $\tilde{O}(\cdot)$ notation.

Table 1: Comparison for different methods within the distributional RL framework. $\mathcal{H}$ represents a subspace of the infinite-dimensional space $\mathcal{F}^\infty$. According to Zanette et al. (2020), the misspecification error $\zeta$, which occurs when the model fails to represent the target, leads to linear regret. Therefore, we can argue that a term with $\Theta = \Omega(\zeta K)$ should be included to capture this effect.

| Algorithm | Regret | Eluder dimension $d_E$ | Bellman Completeness | MDP assumption | Finitely Representable | Exactly Learnable |
|---|---|---|---|---|---|---|
| O-DISCO (Wang et al., 2023) | $\tilde{\mathcal{O}}(\text{poly}(d_E H)\sqrt{K}) + \Theta$ | $\dim_E(\mathcal{H}, \epsilon)$ | distributional BC | discretized reward, small-loss bound | ✗ | ✗ |
| V-EST-LSR (Chen et al., 2024) | $\tilde{\mathcal{O}}(d_E H^2 \sqrt{K}) + \Theta$ [2] | $\dim_E(\mathcal{H}, \epsilon)$ | distributional BC | discretized reward, Lipschitz continuity | ✗ | ✗ |
| SF-LSVI [Ours] | $\tilde{\mathcal{O}}(d_E H^{\frac{3}{2}}\sqrt{K})$ | $\dim_E(\mathcal{F}^N, \epsilon)$ | statistical functional BC | none | ✓ | ✓ |

## 2 RELATED WORK

**Distributional RL.** In classical RL, the Bellman equation, which is based on expected returns, has a closed-form expression. However, it remains unclear whether any statistical functionals of return distribution always have corresponding closed-form expressions. Rowland et al. (2019) introduced the notion of *Bellman Closedness* for collections of statistical functionals that can be updated in a closed form via Bellman update. It was shown that the only Bellman-closed statistical functionals in the discounted setting are the moments $\mathbb{E}_{Z \sim \eta}[Z^k]$. More recently, Marthe et al. (2023) proposed a general framework for distRL where the agent aims to maximize its own utility functionals instead of the expected return, formalizing this property as *Bellman Optimizability*. They further demonstrated that in the undiscounted setting, the only $W_1$-continuous and linear Bellman optimizable statistical functionals are exponential utilities $\frac{1}{\lambda} \log \mathbb{E}_{Z \sim \eta}[\exp(\lambda Z)]$.

In practice, C51 (Bellemare et al., 2017) and QR-DQN (Dabney et al., 2018b) are notable distributional RL algorithms where the convergence guarantees of sampled-based algorithms are proved (Rowland et al., 2018; 2023). Dabney et al. (2018a) expanded the class of policies on arbitrary distortion risk measures by taking the based distribution non-uniformly and improve the sample efficiency from their implicit representation. Cho et al. (2023) highlighted limitations of optimistic exploration in distRL and introduced a randomized exploration method that perturbs the return distribution when the agent selects the next action.

**RL with General Value Function Approximation.** Regret bounds have been extensively studied in online RL, across various domains, such as bandit (Lattimore and Szepesvári, 2020; Abbasi-Yadkori et al., 2011; Russo and Van Roy, 2013), tabular RL (Kakade, 2003; Auer et al., 2008; Osband and Van Roy, 2016; Osband et al., 2019; Jin et al., 2018), and linear function approximation (Jin et al., 2020; Wang et al., 2019; Zanette et al., 2020). In recent years, deep RL has shown significant performance using deep neural networks as function approximators, and there have been attempts to analyze in general function approximation (GVFA) (Jin et al., 2021; Agarwal et al., 2023). Wang et al. (2020) established a provably efficient RL algorithm with general value function approximation based on the eluder dimension $d_E$ (Russo and Van Roy, 2013) and achieves a regret upper bound of $\tilde{O}(\text{poly}(d_E H)\sqrt{K})$. To circumvent the intractability of computing the upper confidence bound, Ishfaq et al. (2021) injected the stochasticity on the training data and get the optimistic value function instead, improving computationally efficiency. Beyond risk-neutral setting, several prior works have shown regret bounds under risk-sensitive objectives (e.g., entropic risk (Fei et al., 2021; Liang and Luo, 2022), CVaR (Bastani et al., 2022)), which partially align with our approach in that they are built on a distribution framework. Liang and Luo (2022) achieved a regret upper bound of $\tilde{O}(\exp(H)\sqrt{|\mathcal{S}|^2|\mathcal{A}|H^2 K})$ and a lower bound of $\Omega(\exp(H)\sqrt{|\mathcal{S}||\mathcal{A}|HK})$ in the tabular setting.

**DistRL with General Value Function Approximation.** Recently, only a few studies have attempted to bridge the gap between distRL and GVFA. Wang et al. (2023) proposed a distRL algorithm, O-DISCO, which enjoys small-loss bound by using a log-likelihood objective. Similarly, Chen et al. (2024) provided a risk-sensitive reinforcement learning framework with static Lipschitz risk measure. While these studies investigate the benefits of distributional framework, their reliance on the distBC assumption raises several potential issues, which we discuss in Section 4.3. In contrast, our approach is grounded in a statistical functional framework. A detailed comparison with other distRL methods is provided in Table 1.

---

[2] In Chen et al. (2024), the regret bound is written as $\tilde{O}(d_E L_\infty(\rho) H \sqrt{K})$, where $L_\infty(\rho)$ represents the Lipschitz constant of the risk measure $\rho$, i.e., $|\rho(Z) - \rho(Z')| \leq L_\infty(\rho)\|F_Z - F_{Z'}\|_\infty$. Since $L_\infty(\rho) \geq H$ in risk-neutral setting, we translate the regret bound into $\tilde{O}(d_E H^2 \sqrt{K})$.

# 3 PRELIMINARIES

**Episodic MDP.** We consider a episodic Markov decision process which is defined as a $\mathcal{M} = (\mathcal{S}, \mathcal{A}, H, \mathbb{P}, r)$ characterized by state space $\mathcal{S}$, action space $\mathcal{A}$, horizon length $H$, transition kernels $\mathbb{P} = \{\mathbb{P}_h\}_{h \in [H]}$, and reward $r = \{r_h\}_{h \in [H]}$ at step $h \in [H]$. The agent interacts with the environment across $K$ episodes. For each $k \in [K]$ and $h \in [H]$, $\mathbb{H}_h^k = (s_1^1, a_1^1, \ldots, s_H^1, a_H^1, \ldots, s_h^k, a_h^k)$ represents the history up to step $h$ at episode $k$. We assume the reward is bounded by $[0, 1]$ and the agent always transit to terminal state $s_{\text{end}}$ at step $H + 1$ with $r_{H+1} = 0$.

**Policy and Value Functions.** A (deterministic) policy $\pi$ is a collection of $H$ functions $\{\pi_h : \mathcal{S} \to \mathcal{A}\}_{h=1}^H$. Given a policy $\pi$, a step $h \in [H]$, and a state-action pair $(s, a) \in \mathcal{S} \times \mathcal{A}$, the $Q$ and $V$-function are defined as $Q_h^\pi(s,a)(: \mathcal{S} \times \mathcal{A} \to \mathbb{R}) := \mathbb{E}_\pi\left[\sum_{h'=h}^H r_{h'}(s_{h'}, a_{h'}) \mid s_h = s, a_h = a\right]$ and $V_h^\pi(s)(: \mathcal{S} \to \mathbb{R}) := \mathbb{E}_\pi\left[\sum_{h'=h}^H r_{h'}(s_{h'}, a_{h'}) \mid s_h = s\right]$.

**Random Variables and Distributions.** For a sample space $\Omega$, we extend the definition of the $Q$-function into a random variable of the cumlative return and its distribution,

$$Z_h^\pi(s,a)(: \mathcal{S} \times \mathcal{A} \times \Omega \to \mathbb{R}) := \sum_{h'=h}^H r_{h'}(s_{h'}, a_{h'}) \mid s_h = s, a_h = a, a_{h'} = \pi_{h'}(s_{h'}),$$

$$\eta_h^\pi(s,a)(: \mathcal{S} \times \mathcal{A} \to \mathscr{P}(\mathbb{R})) := \text{law}(Z_h^\pi(s,a)).$$

Analogously, we extend the definition of $V$-function by introducing a bar notation.

$$\bar{Z}_h^\pi(s)(: \mathcal{S} \times \Omega \to \mathbb{R}) := \sum_{h'=h}^H r_{h'}(s_{h'}, a_{h'}) \mid s_h = s, a_{h'} = \pi_{h'}(s_{h'}),$$

$$\bar{\eta}_h^\pi(s)(: \mathcal{S} \to \mathscr{P}(\mathbb{R})) := \text{law}(\bar{Z}_h^\pi(s)).$$

Note that $\bar{Z}_h^\pi(s) = Z_h^\pi(s, \pi(s))$ and $\bar{\eta}_h^\pi(s) = \eta_h^\pi(s, \pi(s))$. We use $\pi^\star$ to denote an optimal policy (*i.e.*, $\pi_h^*(\cdot|s) = \arg\max_\pi V_h^\pi(s) \; \forall h$) and the corresponding optimal functions as $V_h^\star(s) = V_h^{\pi^\star}(s)$, $Q_h^\star(s,a) = Q_h^{\pi^\star}(s,a)$, $\eta_h^\star(s,a) = \eta_h^{\pi^\star}(s,a)$, and $\bar{\eta}_h^\star(s) = \bar{\eta}_h^{\pi^\star}(s)$. For notational simplicity, we denote the expectation over transition, $[\mathbb{P}_h V_{h+1}^\pi](s,a) = \mathbb{E}_{s' \sim \mathbb{P}_h(\cdot|s,a)} V_{h+1}^\pi(s')$, $[\mathbb{P}_h \bar{Z}_{h+1}^\pi](s,a) = \mathbb{E}_{s' \sim \mathbb{P}_h(\cdot|s,a)} \bar{Z}_{h+1}^\pi(s')$, and $[\mathbb{P}_h \bar{\eta}_{h+1}^\pi](s,a) = \mathbb{E}_{s' \sim \mathbb{P}_h(\cdot|s,a)} \bar{\eta}_{h+1}^\pi(s')$. [3] For brevity, we refer to $\bar{\eta}^\pi$ simply as $\bar{\eta}$.

In the episodic MDP, the agent aims to learn the optimal policy through a fixed number of interactions with the environment across a number of episodes. At the beginning of each episode $k(\in [K])$, the agent starts at the initial state $s_1^k$ and choose a policy $\pi^k$. In step $h(\in [H])$, the agent observes $s_h^k(\in \mathcal{S})$, takes an action $a_h^k(\in \mathcal{A}) \sim \pi_h^k(\cdot|s_h^k)$, receives a reward $r_h(s_h^k, a_h^k)$, and the environment transits to the next state $s_{h+1}^k \sim \mathbb{P}_h(\cdot|s_h^k, a_h^k)$. Finally, we measure the suboptimality of an agent by its regret, which is the accumulated difference between the ground truth optimal and the return received from the interaction. The regret after $K$ episodes is defined as $\text{Reg}(K) = \sum_{k=1}^K V_1^\star(s_1^k) - V_1^{\pi^k}(s_1^k)$.

**Distributional Bellman Optimality Equation.** The distributional Bellman optimality operator $\mathcal{T}_h$ is defined as:

$$(\mathcal{T}_h \eta_{h+1}^\pi)(s,a) := \mathbb{E}_{s' \sim \mathbb{P}_h(\cdot|s,a), a' \sim \pi_h^\star(\cdot|s')}[(\mathcal{B}_{r_h})_\# \eta_{h+1}^\pi(s', a')]$$

where $\mathcal{B}_r : \mathbb{R} \to \mathbb{R}$ is defined by $\mathcal{B}_r(x) = r + x$, and $g_\# \eta \in \mathscr{P}(\mathbb{R})$ is the pushforward of the distribution $\eta$ through $g$ (*i.e.*, $g_\# \eta(A) = \eta(g^{-1}(A))$ for any Borel set $A \subseteq \mathbb{R}$). Then, the optimal return distribution $\eta_h^\star$ satisfies the distributional Bellman optimality equation, $\eta_h^\star(s,a) = (\mathcal{T}_h \eta_{h+1}^\star)(s,a) = (\mathcal{B}_{r_h})_\#[\mathbb{P}_h \eta_{h+1}^\star](s,a)$.

**Additional Notations.** For a given $N$, we denote an $N$-dimensional function class $\mathcal{F}^N := \mathcal{F}^{(1)} \times \cdots \times \mathcal{F}^{(N)} \subseteq \left\{ f = [f^{(1)}, \cdots, f^{(N)}] : \mathcal{S} \times \mathcal{A} \to \mathbb{R}^N \right\}$. Given a dataset $\mathcal{D} = \{(s_t, a_t, [z_t^{(1)}, \ldots, z_t^{(N)}])\}_{t=1}^{|\mathcal{D}|} \subseteq \mathcal{S} \times \mathcal{A} \times \mathbb{R}^N$ and a set of state-action pairs $\mathcal{Z} = \{(s_t, a_t)\}_{t=1}^{|\mathcal{Z}|} \subseteq \mathcal{S} \times \mathcal{A}$, for a function $f : \mathcal{S} \times \mathcal{A} \to \mathbb{R}^N$, we define the norms $\|f^{(n)}\|_\infty, \|f\|_{\infty,1}, \|f\|_{\mathcal{D}}, \|f\|_{\mathcal{Z}}$ as described in Appendix A. For a set of (vector-valued) functions $\mathcal{F}^N \subseteq \{f : \mathcal{S} \times \mathcal{A} \to \mathbb{R}^N\}$, the width function of $(s, a)$ is defined as $w^{(n)}(\mathcal{F}^N, s, a) := \max_{f,g \in \mathcal{F}^N} |f^{(n)}(s,a) - g^{(n)}(s,a)|$.

---

[3]Note that $\mathbb{E}_{s' \sim \mathbb{P}_h(\cdot|s,a)} \bar{\eta}_{h+1}^\pi(s')$ is a mixture distribution.

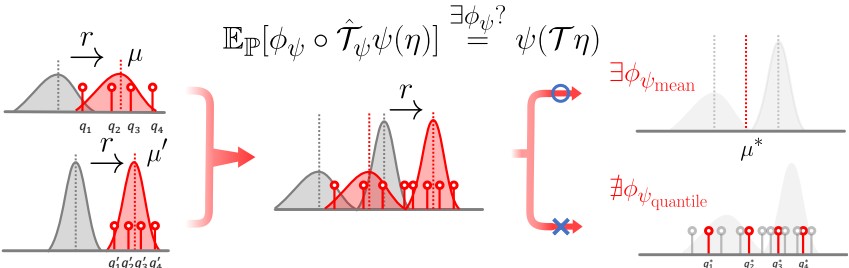

Figure 2: Illustrative representation of sketch-based Bellman updates for a mixture distribution. Instead of updating the distributions directly, each sample distribution is embedded through a sketch $\psi$ (e.g., mean $\mu$, quantile $q_i$). The transformation $\phi_\psi$ is applied to compress the mixture distribution into the same number of parameters, ensuring that the update remains unbiased and prevents information loss.

## 4 STATISTICAL FUNCTIONALS IN DISTRIBUTIONAL RL

In this section, we begin by defining two key concepts commonly used in the distRL framework: the *statistical functional* and the *sketch*. We also illustrate *Bellman closedness*, a crucial property defined by Bellemare et al. (2023). Next, we introduce a novel concept called *Bellman unbiasedness* which complements Bellman closedness and is essential for achieving provable efficiency. As illustrated in Figure 2, quantile functionals cannot be updated in an unbiased manner (as proven in Theorem 4.3), demonstrating that only certain sketches can be updated exactly. Building on this, we show that the moment functional is the only sketch that satisfies both properties, making it unique among the broader class of statistical functionals. Finally, we address the limitations of the previous structural assumption, distributional Bellman Completeness, particularly its tendency to result in linear regret. To overcome this, we introduce a relaxed structural assumption, *statistical functional Bellman Completeness*.

### 4.1 BELLMAN CLOSEDNESS

**Definition 4.1** (Statistical functionals, Sketch; (Bellemare et al., 2023)). *A **statistical functional** is a mapping from a probability distribution to a real value, $\psi : \mathscr{P}(\mathbb{R}) \to \mathbb{R}$. A **sketch** is a vector-valued function, $\psi_{1:N} : \mathscr{P}(\mathbb{R}) \to \mathbb{R}^N$, specified by an $N$-tuple where each component is a statistical functional,*

$$\psi_{1:N}(\cdot) = (\psi_1(\cdot), \cdots , \psi_N(\cdot)).$$

We denote the domain of sketch as $\mathscr{P}_{\psi_{1:N}}(\mathbb{R})$ and its image as $I_{\psi_{1:N}} = \{\psi_{1:N}(\bar{\eta}) : \bar{\eta} \in \mathscr{P}_{\psi_{1:N}}(\mathbb{R})\}$. This can be further extended to state return distribution functions, $\psi_{1:N}(\bar{\eta}) = \left(\psi_{1:N}(\bar{\eta}(s)) : s \in \mathcal{S}\right)$.

**Definition 4.2** (Bellman closedness; (Rowland et al., 2019)). *A sketch $\psi_{1:N}$ is **Bellman closed** if there exists an operator $\mathcal{T}_{\psi_{1:N}} : I^{\mathcal{S}}_{\psi_{1:N}} \to I^{\mathcal{S}}_{\psi_{1:N}}$ such that*

$$\psi_{1:N}(\mathcal{T}\bar{\eta}) = \mathcal{T}_{\psi_{1:N}}\psi_{1:N}(\bar{\eta}) \quad \textit{for all } \bar{\eta} \in \mathscr{P}(\mathbb{R})^{\mathcal{S}},$$

*where the sketch is closed under a distributional Bellman operator $\mathcal{T} : \mathscr{P}(\mathbb{R})^{\mathcal{S}} \to \mathscr{P}(\mathbb{R})^{\mathcal{S}}$.*

Bellman closedness is the property that a sketch are exactly learnable when updates are performed from the infinite-dimensional distribution space to the finite-dimensional embedding space. While the classical Bellman equation implies the existence of Bellman operator for expected values, not all statistical functional have such a corresponding Bellman operator. Precisely, Rowland et al. (2019) showed that the only finite linear statistical functionals that are Bellman closed are those whose linear span is equal to the set of exponential-polynomial functionals. [4]

**Theorem 4.3.** *Quantile functional cannot be Bellman closed under any additional sketch.*

While Rowland et al. (2019) focused on "linear" statistical functionals in defining a sketch, leaving open questions about nonlinear functionals, we extend their work by showing that nonlinear statistical functionals, such as maximum or minimum, can also be Bellman closed. Additionally, while their work implicitly treated quantiles as linear functionals, we clarify this by proving the non-existence of a sketch Bellman operator for quantiles. Further details are provided in Appendix C.1.

---

[4] In discounted setting, a unique solution becomes moments. We've overwritten it for convinience.

Figure 3: **(Left)** Bellman Closedness. **(Right)** Bellman Unbiasedness. The above path represents an ideal distributional Bellman update. Due to the infeasibility of dealing with the infinite-dimensionality of a distribution, the update process should be represented by using a finite-dimensional embedding (sketch) $\psi$. Since the transition kernel $\mathbb{P}$ is unknown, the below path describes that the implementation should sample the next state and update by using $\hat{\mathcal{T}}_\psi$ with the empirical transition kernel $\hat{\mathbb{P}}$. A sketch $\psi$ is Bellman unbiased if $\hat{\mathcal{T}}_\psi \circ \psi$ can unbiasedly estimate $\psi \circ \mathcal{T}$ through some transformation $\phi_\psi$, *i.e.,* $\psi(\mathcal{T}\eta) = \mathbb{E}_{\mathbb{P}}[\phi_\psi \circ \hat{\mathcal{T}}\psi(\eta)]$.

## 4.2 BELLMAN UNBIASEDNESS

While Bellman closedness addresses the infeasibility caused by infinite-dimensionality, another infeasible element that remains unresolved is *the sampling of the next state*. In practice, the agent does not have access to the ground truth transition kernel $\mathbb{P}$. Instead, it relies on the empirical transition kernel $\hat{\mathbb{P}}(\cdot|s,a) = \frac{1}{K}\sum_{k=1}^{K}\mathbf{1}\{s'_k = \cdot \,|s,a\}$ which is derived from $K$ sampled next states. This limitation implies that the operator must be treated as an empirical operator $\hat{\mathcal{T}}_\psi$, rather than $\mathcal{T}_\psi$ (*i.e.,* $\hat{\mathcal{T}}_\psi\psi(\bar\eta) \coloneqq \psi((\mathcal{B}_r)_{\#}[\hat{\mathbb{P}}\bar\eta])$). Therefore, as shown in Figure 3, we naturally introduce a new notion of *Bellman unbiasedness* to unbiasedly estimate the expected distribution $(\mathcal{B}_r)_{\#}\mathbb{E}_{s'\sim\mathbb{P}(\cdot|s,a)}[\bar\eta(s')]$, which is a mixture by transitions, using the sample distribution $(\mathcal{B}_r)_{\#}\bar\eta(s')$.

**Definition 4.4** (Bellman unbiasedness). *A sketch $\psi(=\psi_{1:N})$ is **Bellman unbiased** if, for any $k$ samples, there exists a corresponding vector-valued estimator $\phi_\psi = \phi_\psi(\psi(\eta_1),\cdots,\psi(\eta_k)): (I_\psi^S)^k \to I_\psi^S$ such that the sketch of expected distribution $(\mathcal{B}_r)_{\#}\mathbb{E}_{s'\sim\mathbb{P}(\cdot|s,a)}[\bar\eta(s')]$ can be unbiasedly estimated by $\phi_\psi$ using the $k$ sampled sketches from the sample distribution $(\mathcal{B}_r)_{\#}\bar\eta(s')$, i.e.,*

$$\mathbb{E}_{s'_1,\cdots,s'_k\sim\mathbb{P}(\cdot|s,a)}\left[\phi_\psi\left(\underbrace{\psi\Big((\mathcal{B}_r)_{\#}\bar\eta(s'_1)\Big),\cdots,\psi\Big((\mathcal{B}_r)_{\#}\bar\eta(s'_k)\Big)}_{k \text{ sampled sketches from sample distribution } \hat{\mathcal{T}}_\psi\psi(\bar\eta(s))}\right)\right] = \psi\Big((\mathcal{B}_r)_{\#}\mathbb{E}_{s'\sim\mathbb{P}(\cdot|s,a)}[\bar\eta(s')]\Big).$$

Bellman unbiasedness is another natural definition, similar to Bellman closedness, which takes into account a finite number of samples for the transition. For example, mean-variance sketch is Bellman unbiased as the following unbiased estimator $\phi_{(\mu,\sigma^2)}$ exists for $k$ sampled sketches $\{(\hat\mu_i,\hat\sigma_i^2)\}_{i=1}^{k}$ derived from transition $\mathbb{P}$:

$$(\mu,\sigma^2) = \phi_{(\mu,\sigma^2)}\Big((\hat\mu_1,\hat\sigma_1^2),\cdots,(\hat\mu_k,\hat\sigma_k^2)\Big) = \Big(\frac{1}{k}\sum_{i=1}^{k}\hat\mu_i, \frac{1}{k}\sum_{i=1}^{k}(\hat\mu_i - \frac{1}{k}\sum_{i=1}^{k}\hat\mu_i)^2 + \hat\sigma_i^2\Big)$$

On the other hand, median functional is not Bellman unbiased since there is no unbiased estimator for median. Then, the following question naturally arises;

> *"Which sketches are unbiasedly estimatable under the sketch-based Bellman update?"*

The following lemma answers this question.

**Lemma 4.5.** *Let $F_{\bar\eta}$ be a CDF of the probability distribution $\bar\eta \in \mathscr{P}_\psi(\mathbb{R})^S$. Then a sketch is Bellman unbiased if and only if the sketch is homogeneous over $\mathscr{P}_\psi(\mathbb{R})^S$ of degree $k$, i.e., there exists some vector-valued function $h = h(x_1,\cdots,x_k): \mathcal{X}^k \to \mathbb{R}^N$ such that*

$$\psi(\bar\eta) = \int\cdots\int h(x_1,\cdots,x_k)dF_{\bar\eta}(x_1)\cdots dF_{\bar\eta}(x_k).$$

Lemma 4.5 states that in statistical functional dynamic programming, an unbiasedly estimatable embedding of a distribution can only be structured as functions that are homogeneous of finite degree. For example, the

variance functional is homogeneous of degree 2 (Halmos, 1946). Building on this concept and combining it with the results on Bellman closedness, we prove that even when including a nonlinear statistical functional, the only sketch that can be exactly learned and unbiasedly estimated in a finite-dimensional embedding space is the moment sketch.

**Theorem 4.6.** *The only finite statistical functionals that are both Bellman unbiased and closed are given by the collections of $\psi_1, \ldots, \psi_N$ where its linear span $\{\sum_{n=0}^{N} \alpha_n \psi_n | \alpha_n \in \mathbb{R}, \forall N\}$ is equal to the set of exponential polynomial functionals $\{\eta \to \mathbb{E}_{Z \sim \eta}[Z^l \exp(\lambda Z)] | l = 0, 1, \ldots, L, \lambda \in \mathbb{R}\}$, where $\psi_0$ is the constant functional equal to 1. In discount setting, it is equal to the linear span of the set of moment functionals $\{\eta \to \mathbb{E}_{Z \sim \eta}[Z^l] | l = 0, 1, \ldots, L\}$ for some $L \leq N$.*

Compared to Rowland et al. (2019), we extend the scope beyond linear statistical functionals to include nonlinear statistical functionals, showing the uniqueness of the moment functional. As shown in Figure 1, our theoretical results show that high-order central moments, such as variance or skewness can be exactly learned and unbiasedly estimated. We also reveal that other nonlinear statistical functionals, like median or quantiles, inevitably involve approximation errors due to biased estimations.

Notably, Bellman unbiasedness guarantees a key property related to regret that the sequence of sampled sketches forms a *martingale*. This martingale property is important as it allows us to apply concentration inequalities to bound the probability that the sequence deviates from its expected behavior. Consequently, this enables the construction of a confidence region, a set that contains the true sketch with high probability. ~~In Section 4.3, we emphasize that unbiasedly estimatable sketches mitigate model misspecification, which enable to achieve near-optimal regret in the distRL framework.~~

### 4.3 STATISTICAL FUNCTIONAL BELLMAN COMPLETENESS

We consider distRL with general value function approximation. For successful temporal difference learning, the GVFA framework for classical RL (Wang et al., 2020; Ayoub et al., 2020; Ishfaq et al., 2021) commonly requires the assumption of *Bellman Completeness*. This assumption states that, after applying the Bellman operator, the output lies in the function class $\mathcal{F}$ (*i.e.,* if $f \in \mathcal{F}$, then $\mathcal{T}_h^\pi f \in \mathcal{F}$ for all $\pi, h$).

**DistBC inevitably leads to linear regret.** In seminal works, Wang et al. (2023) and Chen et al. (2024) assumed that the function class $\mathcal{H} \subseteq \{\eta : \mathcal{S} \times \mathcal{A} \to \mathscr{P}([0, H])\}$ follows the *distributional Bellman Completeness* (distBC) assumption (*i.e.,* if $\eta \in \mathcal{H}$, then $\mathcal{T}_h^\pi \eta \in \mathcal{H}$ for all $\pi, h \in [H]$). While this assumption seems natural, constructing a finite-dimensional subspace $\mathcal{H}$ that satisfies distBC is quite challenging. Since the distributional Bellman operator is a composition of translation and mixing distributions for the next state, it implies that the function class $\mathcal{H}$ must be closed under translation and mixture. However, when representing infinite-dimensional distributions using a finite number of parameters or statistical functionals, it is not trivial to ensure that a mixture of distributions can also be represented with the same number of parameters or statistical functionals. For example, while a single Gaussian distribution can be represented using two parameters $(\mu, \sigma^2)$, a mixture of $K$ Gaussians generally requires $2K$ parameters for exact representation.

To address the issue of closedness under mixture, both previous studies assumed a discretized reward MDP where all outcomes of the return distribution are discretized into a uniform grid of finite points. Unfortunately, the approximation error introduced by this discretization is not negligible with respect to regret. This is because the approximation error is, in fact, a form of model misspecification – an error arising when the model fails to represent the target – which makes it difficult to achieve provable efficiency.

**Definition 4.7** (Model Misspecification in distBC). *For a given distribution class $\mathcal{H}$, which is a finite-dimensional subspace of the space of all distribution $\mathcal{F}^\infty$, we define $\zeta$ as the **model misspecification***

$$\zeta := \inf_{f_{\bar{\eta}} \in \mathcal{H}} \sup_{(s,a) \in \mathcal{S} \times \mathcal{A}} \|f_{\bar{\eta}}(s, a) - (\mathcal{B}_{r_h})_{\#}[\mathbb{P}_h \bar{\eta}](s, a)\|$$

*for any $\bar{\eta} : \mathcal{S} \to \mathscr{P}([0, H])$ and $h \in [H]$.*

Note that $\zeta$ is strictly positive unless the function approximator $f_{\bar{\eta}}$ can represent any distribution in the finite-dimensional subspace $\mathcal{H}$ generated by translation and mixture. In the classical linear bandit setting (Zanette et al., 2020), a lower bound with model misspecification $\zeta$ is known to yield linear regret $\Omega(\zeta K)$. Therefore, redefining Bellman Completeness within the infinite-dimensional distribution space is not appropriate, as it either imposes overly restrictive constraints on the MDP structure or leads to linear regret.

To circumvent model misspecification, we revisit the concept of distBC through the statistical functional lens. We propose a novel framework that matches a finite number of statistical functionals to the target, rather than attempting to represent the entire distribution. As a natural extension, our approach takes a tuple of function class $\mathcal{F}^N \subseteq \{f : \mathcal{S} \times \mathcal{A} \to \mathbb{R}^N\}$ as input to represent $N$ moments of the distribution. Building on this, we assume that for any $\bar{\eta} : \mathcal{S} \to \mathscr{P}([0, H])$, the sketch of target function lies in the function class $\mathcal{F}^N$.

**Assumption 4.8** (Statistical Functional Bellman Completeness). *For any distribution $\bar{\eta} : \mathcal{S} \to \mathscr{P}([0, H])$ and $h \in [H]$, there exists $f_{\bar{\eta}} \in \mathcal{F}^N$ which satisfies*

$$f_{\bar{\eta}}(s, a) = \psi_{1:N}\big((\mathcal{B}_{r_h})_\# [\mathbb{P}_h \bar{\eta}](s, a)\big) \quad \forall (s, a) \in \mathcal{S} \times \mathcal{A}$$

## 5 SF-LSVI: STATISTICAL FUNCTIONAL LEAST SQUARE VALUE ITERATION

In this section, we propose SF-LSVI for the distRL framework with general value function approximation. Leveraging the result from Theorem 4.6, we introduce a moment least square regression. This approach allows us to capture a finite set of moment information from the distribution, which can be unbiasedly estimated. Unlike previous works (Wang et al., 2023; Chen et al., 2024), which rely on estimating the entire distribution in infinite-dimensional spaces, our method enables to estimate unbiasedly in finite-dimensional embedding spaces, effectively avoiding model misspecification.

---

**Algorithm 1** Statistical Functional Least Square Value Iteration (SF-LSVI($\delta$))

---

**Input:** failure probability $\delta \in (0, 1)$ and the number of episodes $K$
1: **for** episode $k = 1, 2, \ldots, K$ **do**
2:     Receive initial state $s_1^k$
3:     Initialize $\psi_{1:N}(\bar{\eta}_{H+1}^k(\cdot)) \leftarrow \mathbf{0}^N$
4:     **for** step $h = H, H - 1, \ldots, 1$ **do**
5:         $\mathcal{D}_h^k \leftarrow \left\{ s_{h'}^\tau, a_{h'}^\tau, \psi_{1:N}\big((\mathcal{B}_{r_{h'}^\tau})_\# \bar{\eta}_{h+1}^k(s_{h'+1}^\tau)\big) \right\}_{(\tau, h') \in [k-1] \times [H]}$     // Data collection
6:         $\tilde{f}_{h, \bar{\eta}}^k \leftarrow \arg\min_{f \in \mathcal{F}^N} \|f\|_{\mathcal{D}_h^k}$     // Distribution Estimation
7:         $b_h^k(\cdot, \cdot) \leftarrow w^{(1)}((\mathcal{F}^N)_h^k, \cdot, \cdot)$
8:         $Q_h^k(\cdot, \cdot) \leftarrow \min\{(\tilde{f}_{h, \bar{\eta}}^k)^{(1)}(\cdot, \cdot) + b_h^k(\cdot, \cdot), H\}$
9:         $\pi_h^k(\cdot) = \arg\max_{a \in \mathcal{A}} Q_h^k(\cdot, a), V_h^k(\cdot) = Q_h^k(\cdot, \pi_h^k(\cdot))$     // Optimistic planning
10:        $\psi_1\big(\eta_h^k(\cdot, \cdot)\big) \leftarrow Q_h^k(\cdot, \cdot), \ \psi_{2:N}\big(\eta_h^k(\cdot, \cdot)\big) \leftarrow \Big( \min\{(\tilde{f}_{h, \bar{\eta}}^k)^{(n)}(\cdot, \cdot), H\}\Big)_{n \in [2:N]}$
11:        $\psi_1\big(\bar{\eta}_h^k(\cdot)\big) \leftarrow V_h^k(\cdot), \ \psi_{2:N}\big(\bar{\eta}_h^k(\cdot)\big) \leftarrow \psi_{1:N}\big(\eta_h^k(\cdot, \pi_h^k(\cdot))\big)_{n \in [2:N]}$
12:     **for** $h = 1, 2, \ldots, H$ **do**
13:         Take action $a_h^k \leftarrow \pi_h^k(s_h^k)$
14:         Observe reward $r_h^k(s_h^k, a_h^k)$ and get next state $s_{h+1}^k$.

---

**Overview.** At the beginning of episode $k \in [K]$, we maintain all previous samples $\{(s_{h'}^\tau, a_{h'}^\tau, r_{h'}^\tau)\}_{(\tau, h') \in [k-1] \times [H]}$ and initialize a sketch $\psi_{1:N}(\bar{\eta}_{H+1}^k(\cdot)) = \mathbf{0}^N$. For each step $h = H, \ldots, 1$, we compute the normalized sample moments of target distribution $\{(\mathcal{B}_{r_{h'}^\tau})_\# \bar{\eta}_{h+1}^k(s_{h'+1}^\tau)\}_{h' \in [H]}$ with the help of binomial theorem,

$$\psi_n\big((\mathcal{B}_{r_{h'}^\tau})_\# \bar{\eta}_h(s_{h'+1}^\tau)\big) := \frac{\mathbb{E}[(\bar{Z}_{h+1}^k(s_{h'+1}^\tau) + r_{h'}^\tau)^n]}{H^{n-1}} = \frac{\sum_{n'=0}^n H^{n'} \psi_{n'}\big(\bar{\eta}_h^k(s_{h'+1}^\tau)\big)(r_{h'}^\tau)^{n-n'}}{H^{n-1}}$$

where $\bar{Z}_h^k$ and $\bar{\eta}_h^k$ denote the random variable and the distribution of returns in the $k$-th episode, respectively. Then we iteratively solve the $N$-moment least square regression

$$\tilde{f}_{h, \bar{\eta}}^k \leftarrow \arg\min_{f \in \mathcal{F}} \sum_{\tau=1}^{k-1} \sum_{h'=1}^H \Big( \sum_{n=1}^N f^{(n)}(s_{h'}^\tau, a_{h'}^\tau) - \psi_n\big((\mathcal{B}_{r_{h'}^\tau})_\# \bar{\eta}_{h+1}^k(s_{h'+1}^\tau)\big) \Big)^2$$

based on the dataset $\mathcal{D}_h^k$ which contains the sketch of temporal target $\psi_{1:N}\big((\mathcal{B}_{r_{h'}^\tau})_\# \bar{\eta}_{h+1}^k(s_{h'+1}^\tau)\big)$. Then we define $Q_h^k(\cdot, \cdot) = \min\{(\tilde{f}_{h, \bar{\eta}}^k)^{(1)}(\cdot, \cdot) + b_h^k(\cdot, \cdot), H\}$ and choose the greedy policy $\pi_h^k(\cdot)$ with respect to $Q_h^k$. Next, we update all $N$ normalized moments of $Q$-distribution $\psi_{1:N}\big(\eta_k^h(\cdot, \cdot)\big)$ and $V$-distribution $\psi_{1:N}\big(\bar{\eta}_k^h(\cdot)\big)$. We repeat the procedure until all the $K$ episodes are completed.

# 6 THEORETICAL ANALYSIS

In this section, we provide the theoretical guarantees for `SF-LSVI` under Assumption 4.8. By applying proof techniques from Wang et al. (2020) and extending the result through a statistical functional lens, we generalize the concept of *eluder dimension* (Russo and Van Roy, 2013) to vector-valued function. This complexity measure has been widely used in RL literatures (Ayoub et al., 2020; Wang et al., 2020; Jin et al., 2020) to quantify the complexity of learning with function approximators.

**Definition 6.1** ($\epsilon$-dependent, $\epsilon$-independent, Eluder dimension for vector-valued function). *Let $\epsilon \geq 0$ and $\mathcal{Z} = \{(s_i, a_i)\}_{i=1}^n \subseteq \mathcal{S} \times \mathcal{A}$ be a sequence of state-action pairs.*

- *A state-action pair $(s, a) \in \mathcal{S} \times \mathcal{A}$ is $\boldsymbol{\epsilon}$-**dependent** on $\mathcal{Z}$ with respect to $\mathcal{F}^N$ if $\|f - g\|_{\mathcal{Z}} \leq \epsilon$ for any vector-valued function $f, g \in \mathcal{F}^N$, then $|f^{(1)}(s, a) - g^{(1)}(s, a)| \leq \epsilon$.*

- *An $(s, a)$ is $\boldsymbol{\epsilon}$-**independent** on $\mathcal{Z}$ with respect to $\mathcal{F}^N$ if $(s, a)$ is not $\epsilon$-dependent on $\mathcal{Z}$.*

- *The $\boldsymbol{\epsilon}$-**eluder dimension** $dim_E(\mathcal{F}^N, \epsilon)$ of a vector-valued function class $\mathcal{F}^N$ is the length of the longest sequence of elements in $\mathcal{S} \times \mathcal{A}$ such that, for some $\epsilon' \geq \epsilon$, every element is $\epsilon'$-independent on its predecessors.*

We assume that the function class $\mathcal{F}^N$ and state-action space $\mathcal{S} \times \mathcal{A}$ have bounded covering numbers.

**Assumption 6.2** (Covering number). *For any $\epsilon > 0$, the following holds:*

- *there exists an $\epsilon$-cover $\mathcal{C}(\mathcal{F}^N, \epsilon) \subseteq \mathcal{F}^N$ with size $|\mathcal{C}(\mathcal{F}^N, \epsilon)| \leq \mathcal{N}(\mathcal{F}^N, \epsilon)$, such that for any $g \in \mathcal{F}^N$, there exists $g' \in \mathcal{C}(\mathcal{F}^N, \epsilon)$ with $\|g - g'\|_{\infty,1} \leq \epsilon$.*

- *there exists an $\epsilon$-cover $\mathcal{C}(\mathcal{S} \times \mathcal{A}, \epsilon)$ with size $|\mathcal{C}(\mathcal{S} \times \mathcal{A}, \epsilon)| \leq \mathcal{N}(\mathcal{S} \times \mathcal{A}, \epsilon)$, such that for any $(s, a) \in \mathcal{S} \times \mathcal{A}$, there exists $(s', a') \in \mathcal{C}(\mathcal{S} \times \mathcal{A}, \epsilon)$ with $\max_{f \in \mathcal{F}} |f(s, a) - f(s', a')| \leq \epsilon$*

The following two lemmas give confidence bounds on the sum of the $l_2$ norms of all normalized moments.

**Lemma 6.3** (Single Step Optimization Error). *Consider a fixed $k \in [K]$ and a fixed $h \in [H]$. Let $\mathcal{Z}_h^k = \{(s_h^\tau, a_h^\tau)\}_{\tau \in [k-1]}$ and $\mathcal{D}_{h,\bar{\eta}}^k = \left\{ \left( s_h^\tau, a_h^\tau, \psi_{1:N}\left( (\mathcal{B}_{r_h})_\# \bar{\eta}(s_{h+1}^\tau) \right) \right) \right\}_{\tau \in [k-1]}$ for any $\bar{\eta} : \mathcal{S} \to \mathscr{P}([0, H])$. Define $\tilde{f}_{h,\bar{\eta}}^k = \arg\min_{f \in \mathcal{F}^N} \|f\|_{\mathcal{D}_{h,\bar{\eta}}^k}^2$. For any $\bar{\eta}$ and $\delta \in (0, 1)$, there is an event $\mathcal{E}(\bar{\eta}, \delta)$ such that conditioned on $\mathcal{E}(\bar{\eta}, \delta)$, with probability at least $1 - \delta$, for any $\bar{\eta}' : \mathcal{S} \to \mathscr{P}([0, H])$ with $\|\psi_{1:N}(\bar{\eta}') - \psi_{1:N}(\bar{\eta})\|_{\infty,1} \leq 1/T$, we have*

$$\left\| \tilde{f}_{h,\bar{\eta}'}(\cdot, \cdot) - \psi_{1:N}\left( (\mathcal{B}_{r(\cdot,\cdot)})_\# [\mathbb{P}\bar{\eta}'](\cdot, \cdot) \right) \right\|_{\mathcal{Z}_h^k} \leq c' \left( N^{\frac{1}{2}} H \sqrt{\log(1/\delta) + \log \mathcal{N}(\mathcal{F}^N, 1/T)} \right)$$

*for some constant $c' > 0$.*

Due to the definition of Bellman unbiasedness, we note that the moment sketch has a corresponding vector-valued estimator $\phi_{\psi_{1:N}}$ as the identity function. This leads to a concentration results as the sequence of sampled sketches forms a martingale with respect to the filtration $\mathbb{F}_h^\tau$ induced by the history of $\{(s_h^\tau, a_h^\tau)\}_{\tau \in [k-1]}$ (i.e., $\mathbb{E}\left[ \psi_{1:N}\left( (\mathcal{B}_{r_h})_\# \bar{\eta}(s_h^\tau) \right) \Big| \mathbb{F}_h^\tau \right] = \psi_{1:N}\left( (\mathcal{B}_{r_h})_\# [\mathbb{P}_h \bar{\eta}](s_h^\tau, a_h^\tau) \right)$).

Another notable aspect of Lemma 6.3 is the use of normalized moments $\mathbb{E}[Z^n]/H^{n-1}$ instead of raw moments $\mathbb{E}[Z^n]$. This normalization reduces the size of the confidence region from $O(H^N)$ to $O(\sqrt{N})$. This adjustment is similar to scaling the objective functions in multi-objective optimization to treat each objective equally, which prevents the model from disproportionally favoring objectives with larger scales.

**Lemma 6.4** (Confidence Region). *Let $(\mathcal{F}^N)_h^k = \{f \in \mathcal{F}^N | \|f - \tilde{f}_{h,\bar{\eta}}^k\|_{\mathcal{Z}_h^k}^2 \leq \beta(\mathcal{F}^N, \delta)\}$, where*

$$\beta(\mathcal{F}^N, \delta) \geq c' \cdot NH^2(\log(T/\delta) + \log \mathcal{N}(\mathcal{F}^N, 1/T))$$

*for some constant $c' > 0$. Then with probability at least $1 - \delta/2$, for all $k, h \in [K] \times [H]$, we have*

$$\psi_{1:N}\left( (\mathcal{B}_{r_h(\cdot,\cdot)})_\# [\mathbb{P}_h \bar{\eta}_{h+1}^k](\cdot, \cdot) \right) \in (\mathcal{F}^N)_h^k.$$

Lemma 6.4 guarantees that the sequence of moments from the target distribution $\psi_{1:N}\left((\mathcal{B}_{r_h(\cdot,\cdot)})_\#[\mathbb{P}_h\bar{\eta}_{h+1}^k](\cdot,\cdot)\right)$ lies in the confidence region $(\mathcal{F}^N)_h^k$ with high probability. Supported by the aforementioned lemma, we can further guarantee that all $Q$-functions are optimistically estimated with high probability, leading to the derivation of our final result.

**Theorem 6.5.** *Under Assumption 4.8, with probability at least $1 - \delta$, SF-LSVI achieves a regret bound of*

$$Reg(K) \leq 2H\,dim_E(\mathcal{F}^N, 1/T) + 4H\sqrt{KH\log(1/\delta)}.$$

Under weaker structural assumptions, we show that `SF-LSVI` enjoys a near-optimal regret bound of order $\tilde{O}(d_E H^{\frac{3}{2}}\sqrt{K})$, which is $\sqrt{H}$ better than the state-of-the-art distRL algorithm `V-EST-LSR` (Chen et al., 2024). In the linear MDP setting, we have $d_E = \tilde{O}(d)$ and thus `SF-LSVI` achieves a tight regret bound as $\tilde{O}(\sqrt{d^2 H^3 K})$, which matches the lower bound $\Omega(\sqrt{d^2 H^3 K})$ (Zhou et al., 2021). Our analysis highlights two main technical novelties that significantly reduce the order of regret in distRL framework:

1. We remove the dependency of $\beta(\mathcal{F}^N, \delta) = \tilde{O}(NH^2 \log \mathcal{N}(\mathcal{F}^N, 1/T))$ in our final results, compared to previous works where the regret bound took the form $\mathrm{Reg}(K) = \tilde{O}(d_E H \sqrt{\beta(\mathcal{F}^N, 1/\delta)K})$ (Osband et al., 2019; Wang et al., 2020) (See Appendix D.4). This refinement ensures that the regret bound depends only on the pre-defined function class, rather than the number of extracted moment.

2. As shown in Table 1, we define the eluder dimension $d_E$ in a finite-dimensional embedding space $\mathcal{F}^N$, while other methods rely on an infinite-dimensional distribution space $\mathcal{H} \subseteq \mathcal{F}^\infty$.

# 7 CONCLUSIONS

To overcome the infeasibility of dealing with the infinite-dimensionality of a distribution, we analyzed the approximation error (model misspecification) inherent in distribution-based updates. We introduced a new concept of Bellman unbiasedness, which enables to exactly learn a distribution with a sample efficient manner. Based on this, we present a provably efficient online distRL algorithm, `SF-LSVI`, with general value function approximation. Notably, our algorithm achieves a near-optimal regret bound of $\tilde{O}(d_E H^{\frac{3}{2}}\sqrt{K})$, matching the tightest upper bound achieved by non-distributional frameworks (Zhou et al., 2021; He et al., 2023).

One might wonder whether representing a distribution with a finite number of moments still introduces approximation errors. Indeed, a reconstructed distribution using only finite information cannot perfectly match the ground truth within infinite-dimensional distribution space. However, when measuring discrepancies between two distributions within a finite-dimensional representation space, moment sketches provide a unique characteristic of being the only representation that introduces no approximation error for these distributions. Leveraging this property, an interesting future direction would be to reformulate regret in terms of moment discrepancies rather than expected returns and to demonstrate the sample efficiency of distRL. It is hoped that this work triggers new insights and ideas for future research in establishing the provable efficiency in practical distRL settings.

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

# APPENDIX

## A NOTATION

Table 2: Table of notation

| Notation | Description |
|---|---|
| $\mathcal{S}$ | state space of size $S$ |
| $\mathcal{A}$ | action space of size $A$ |
| $H$ | horizon length of one episode |
| $T$ | number of episodes |
| $r_h(s,a)$ | reward of $(s,a)$ at step $h$ |
| $\mathbb{P}_h(s'|s,a)$ | probability transition of $(s,a)$ to $s'$ at step $h$ |
| $\mathbb{H}_h^k$ | history up to step $h$, episode $k$ |
| $N$ | number of statistical functionals |
| $Q_h^\pi(s,a)$ | $Q$-function of a given policy $\pi$ at step $h$ |
| $V_h^\pi(s)$ | $V$-function of a given policy $\pi$ at step $h$ |
| $Z_h^\pi(s,a)$ | random variable of cumulative return $\sum_{h'=h}^H r_{h'}(s_{h'},a_{h'})\,|s_h=s, a_h=a, a_{h'}=\pi_{h'}(s_{h'})$ |
| $\bar{Z}_h^\pi(s)$ | random variable of cumulative return $\sum_{h'=h}^H r_{h'}(s_{h'},a_{h'})\,|s_h=s, a_{h'}=\pi_{h'}(s_{h'})$ |
| $\eta_h^\pi(s,a)$ | probability distribution of $Z_h^\pi(s,a)$ |
| $\bar{\eta}_h^\pi(s)$ | probability distribution of $\bar{Z}_h^\pi(s)$ |
| $[\mathbb{P}_h(\cdot)]$ | expectation over transition $[\mathbb{P}_h(\cdot)] = \mathbb{E}_{s'\sim\mathbb{P}_h}(\cdot)$ |
| $(\mathcal{B}_r)_\#$ | pushforward of the distribution through $\mathcal{B}_r(x) := r + x$ |
| $f^{(n)}$ | $n$-th element of $N$-dimensional vector $f$ |
| $\|f\|_\infty$ | max norm of $f: X \to \mathbb{R}$ defined as $\|f\|_\infty := \max_{x\in X}|f^{(n)}(x)|$ |
| $\|f\|_{\infty,1}$ | $l_1$-norm of max norm of $f: X \to \mathbb{R}$ defined as $\|f\|_{\infty,1} := \sum_{n=1}^N \max_{x\in X}|f^{(n)}(x)|$ |
| $\mathcal{F}^N$ | a function class of $N$-dimensional embedding space |
| $\mathcal{Z}$ | a set of state-action pairs $\mathcal{Z} := \{(s_t,a_t)\}_{t=1}^{|\mathcal{Z}|}$ |
| $\mathcal{D}$ | a dataset $\mathcal{D} := \{(s_t,a_t,[d_t^{(1)},\cdots,d_t^{(N)}])\}_{t=1}^{|\mathcal{D}|}$ |
| $\|f\|_{\mathcal{Z}}^2$ | for $f: \mathcal{S}\times\mathcal{A}\to\mathbb{R}$, define $\|f\|_{\mathcal{Z}}^2 := \sum_{n=1}^N \sum_{(s,a)\in\mathcal{Z}}(f^{(n)}(s_t,a_t))^2$ |
| $\|f\|_{\mathcal{D}}^2$ | for $f: \mathcal{S}\times\mathcal{A}\to\mathbb{R}$, define $\|f\|_{\mathcal{D}}^2 := \sum_{n=1}^N \sum_{t=1}^{\mathcal{D}}(f^{(n)}(s_t,a_t)-d_t^{(n)})^2$ |
| $w^{(n)}(\mathcal{F}^N,s,a)$ | width function of $(s,a)$ defined as $w^{(n)}(\mathcal{F}^N,s,a) := \max_{f,g\in\mathcal{F}^N}|f^{(n)}(s,a)-g^{(n)}(s,a)|$ |
| $\tilde{f}_{h,\bar{\eta}}^k$ | a solution of moment least square regression, defined as $\tilde{f}_{h,\bar{\eta}}^k := \arg\min_{f\in\mathcal{F}^N}\|f\|_{\mathcal{D}_h^k}$ |
| $f_{\bar{\eta}}$ | a target sketch of distribution $\bar{\eta}$, defined as $f_{\bar{\eta}} := \psi_{1:N}((\mathcal{B}_r)_\#[\mathbb{P}_h\bar{\eta}])$ |
| $(\mathcal{F}^N)_h^k$ | a confidence region at step $h$, episode $k$, defined as $(\mathcal{F}^N)_h^k := \{f\in\mathcal{F}^N\,|\,\|f-\tilde{f}_{h,\bar{\eta}}^k\|_{\mathcal{Z}_h^k}^2 \le \beta(\mathcal{F}^N,\delta)\}$ |
| $\psi(\bar{\eta})$ | a statistical functional $\mathscr{P}_\psi(\mathbb{R})^{\mathcal{S}} \to \mathbb{R}^S$ |
| $\psi_{1:N}(\bar{\eta})$ | a $N-$collection of statistical functional $\mathscr{P}_{\psi_{1:N}}(\mathbb{R})^{\mathcal{S}} \to \mathbb{R}^{N\times S}$ |
| $\mathscr{P}_{\psi_{1:N}}(\mathbb{R})$ | a domain of sketch $\psi_{1:N}$ |
| $I_{\psi_{1:N}}$ | an image of sketch $\psi_{1:N}$ |
| $\mathcal{T}$ | distributional Bellman operator, defined as $\mathcal{T}\bar{\eta} := (\mathcal{B}_r)_\#[\mathbb{P}\bar{\eta}]$ |
| $\mathcal{T}_\psi$ | sketch Bellman operator w.r.t $\psi$, defined as $\mathcal{T}_\psi\psi(\bar{\eta}) := \psi\big((\mathcal{B}_r)_\#[\mathbb{P}\bar{\eta}]\big)$ |
| $\hat{\mathcal{T}}_\psi$ | empirical sketch Bellman operator w.r.t $\psi$, defined as $\hat{\mathcal{T}}_\psi\psi(\bar{\eta}) := \psi\big((\mathcal{B}_r)_\#[\hat{\mathbb{P}}\bar{\eta}]\big)$ |
| $\mathcal{N}(\mathcal{F}^N,\epsilon)$ | covering number of $\mathcal{F}^N$ w.r.t the $\epsilon-$ball |
| $\dim_E(\mathcal{F}^N,\epsilon)$ | eluder dimension of $\mathcal{F}^N$ w.r.t $\epsilon$ |

## B    TECHNICAL REMARKS

For an optimistic planning in Line 7 of pseudocode 1, we define the bonus function as the width function $b_h^k(s,a) := w_h^k((\mathcal{F}^N)_h^k, s, a)$ where $(\mathcal{F}^N)_h^k$ denotes a confidence region at step $h$, episode $k$. When $\mathcal{F}$ is a linear function class, the width function can be evaluated by simply computing the maximal distance of weight vector. For a general function class $\mathcal{F}$, computing the width function requires to solve a set-constrained optimization problem, which is known as NP-hard Dann et al. (2018). However, a width function is computed simply for optimistic exploration, and approximation errors are known to have a small effect on regret Abbasi-Yadkori et al. (2011).

## C    RELATED WORK AND DISCUSSION

### C.1    TECHNICAL CLARIFICATIONS ON LINEARITY ASSUMPTION IN EXISTING RESULTS

**Bellman Closedness and Linearity.**    Rowland et al. (2019) proved that quantile functional is not Bellman closed by providing a specific counterexample. However, their discussion based on counterexamples can be generalized as it assumes that the sketch Bellman operator for the quantile functional needs to be linear.

They consider an discounted MDP with initial state $s_0$ with single action $a$, which transits to one of two terminal states $s_1, s_2$ with equal probability. Letting no reward at state $s_0$, $\text{Unif}([0,1])$ at state $s_1$, and $\text{Unif}([1/K, 1+1/K])$ at state $s_2$, the return distribution at state $s_0$ is computed as mixture $\frac{1}{2}\text{Unif}([0,\gamma]) + \frac{1}{2}\text{Unif}([\gamma/K, \gamma+\gamma/K])$. Then the $\frac{1}{2K}$−quantile at state $s_0$ is $\frac{\gamma}{K}$. They proposed a counterexample where each quantile distribution of state $s_1, s_2$ is represented as $\frac{1}{K}\sum_{k=1}^{K}\delta_{\frac{2k-1}{K}}$ and $\frac{1}{K}\sum_{k=1}^{K}\delta_{\frac{2k+1}{K}}$ respectively, the $\frac{1}{2K}$−quantile of state $s_0$ is $\psi_{q_{2K}}\left(\frac{1}{2K}\sum_{k=1}^{K}\delta_{\frac{\gamma(2k-1)}{K}}+\delta_{\frac{\gamma(2k+1)}{K}}\right) = \frac{3\gamma}{2K}$. However, this example does not consider that the mixture of quantiles is not a quantile of the mixture distribution (i.e., $\psi_q(\lambda\eta_1+(1-\lambda)\eta_2) \neq \lambda\psi_q(\eta_1)+(1-\lambda)\psi_q(\eta_2)$), due to the nonlinearity of the quantile functional. Therefore, this does not present a valid counterexample to prove that quantile functionals are not Bellman closed.

**Bellman Optimizability and Linearity.**    Marthe et al. (2023) proposed the notion of Bellman optimizable statistical functional which redefine the Bellman update by planning with respect to statistical functionals rather than expected returns. They proved that $W_1$-continuous Bellman Optimizable statistical functionals are characterized by exponential utilities $\frac{1}{\lambda}\log\mathbb{E}_{Z\sim\eta}[\exp(\lambda Z)]$. However, their proof requires some technical clarification regarding the assumption that such statistical functionals are linear.

To illustrate, they define a statistical functional $\psi_f$ and consider two probability distributions $\eta_1 = \frac{1}{2}(\delta_0+\delta_h)$ and $\eta_2 = \delta_{\phi(h)}$ where $\phi(h) = f^{-1}\left(\frac{1}{2}(f(0)+f(h))\right)$. Using the translation property, they lead $\psi_f(\eta_1) = \psi_f(\eta_2)$ to $\frac{1}{2}(f(x)+f(x+h)) = f(x+\phi(h))$ for all $x \in \mathbb{R}$. However, this equality $\psi_f\left(\frac{1}{2}(\delta_x+\delta_{x+h})\right) = \frac{1}{2}(f(x)+f(x+h))$ holds only if $\psi_f$ is linear, which is not necessarily a valid assumption for all statistical functionals.

### C.2    EXISTENCE OF NONLINEAR BELLMAN CLOSED SKETCH.

The previous two examples may not have considered the possibility that the sketch Bellman operator might not necessarily be linear. However, some statistical functionals are Bellman-closed even if they are nonlinear, so it is open question whether there is a nonlinear sketch Bellman operator that makes the quantile functional Bellman-closed. In this section, we present examples of maximum and minimum functionals that are Bellman-closed, despite being nonlinear.

In a nutshell, consider the maximum of return distribution at state $s_1, s_2$ is $\gamma, \gamma+\gamma/K$ respectively. Beyond linearity, the maximum of return distribution at state $s_0$ can be computed by taking the maximum of these values;

$$\max(\max(\bar{\eta}(s_1)), \max(\bar{\eta}(s_2))) = \max(\gamma, \gamma+\gamma/K) = \gamma+\gamma/K$$

which produces the desired result. This implies the existence of a nonlinear sketch that is Bellman closed. More precisely, by defining $\max_{s'\sim\mathbb{P}(\cdot|s,a)}$ and $\min_{s'\sim\mathbb{P}(\cdot|s,a)}$ as the maximum and minimum of the sampled sketch $\psi\left((\mathcal{B}_r)_{\#}\bar{\eta}(s')\right)$ with the distribution $\mathbb{P}(\cdot|s,a)$, we can derive the sketch Bellman operator for maximum and minimum functionals as follows;

- $\mathcal{T}_{\psi_{\max}}\Big(\psi_{\max}(\bar{\eta}(s))\Big) = \max_{s' \sim \mathbb{P}(\cdot|s,a)}\Big(\psi_{\max}\big((\mathcal{B}_r)_{\#}\bar{\eta}(s')\big)\Big)$

- $\mathcal{T}_{\psi_{\min}}\Big(\psi_{\min}(\bar{\eta}(s))\Big) = \min_{s' \sim \mathbb{P}(\cdot|s,a)}\Big(\psi_{\min}\big((\mathcal{B}_r)_{\#}\bar{\eta}(s')\big)\Big).$

## C.3 NON-EXISTENCE OF SKETCH BELLMAN OPERATOR FOR QUANTILE FUNCTIONAL

In this section, we prove that quantile functional cannot be Bellman closed under any additional sketch. First we introduce the definition of *mixture-consistent*, which is the property that the sketch of a mixture can be computed using only the sketch of the distribution of each component.

**Definition C.1** (mixture-consistent). *A sketch $\psi$ is **mixture-consistent** if for any $\lambda \in [0,1]$ and any distributions $\eta_1, \eta_2 \in \mathscr{P}_\psi(\mathbb{R})$, there exists a corresponding function $h_\psi$ such that*

$$\psi(\lambda\eta_1 + (1-\lambda)\eta_2) = h_\psi\Big(\psi(\eta_1), \psi(\eta_2), \lambda\Big).$$

Next, we will provide some examples of determining whether a sketch is mixture-consistent or not.

**Example 1.** Every moment or exponential polynomial functional is mixture-consistent.

*Proof.* For any $n \in [N]$ and $\lambda \in \mathbb{C}$,

$$\mathbb{E}_{Z \sim \lambda\eta_1 + (1-\lambda)\eta_2}[Z^n \exp(\lambda Z)] = \lambda\mathbb{E}_{Z \sim \eta_1}[Z^n \exp(\lambda Z)] + (1-\lambda)\mathbb{E}_{Z \sim \eta_2}[Z^n \exp(\lambda Z)].$$

∎

**Example 2.** Variance functional is not mixture-consistent.

*Proof.* Let $\lambda = \frac{1}{2}$ and $Z, Y$ be the random variables where $Z \sim \frac{1}{2}\delta_0 + \frac{1}{2}\delta_2$ and $Y \sim \frac{1}{2}\delta_k + \frac{1}{2}\delta_{k+2}$. Then, $\mathrm{Var}(Z) = \mathrm{Var}(Y) = 1$. While RHS is constant for any $k$, LHS is not a constant for any $k$, i.e.,

$$\mathrm{Var}_{X \sim \frac{1}{2}(\frac{1}{2}\delta_0 + \frac{1}{2}\delta_2) + \frac{1}{2}(\frac{1}{2}\delta_k + \frac{1}{2}\delta_{k+2})}(X) = \frac{1}{4}(k^2 + 5).$$

∎

While variance functional is not mixture consistent by itself, it can be mixture consistent with another statistical functional, the mean.

**Example 3.** Variance functional is mixture-consistent under mean functional.

*Proof.* Notice that mean functional is mixture-consistent. We need to show that variance functional is mixture-consistent under mean functional.

$$\begin{aligned}
&\mathrm{Var}_{Z \sim \lambda\eta_1 + (1-\lambda)\eta_2}[Z] \\
&= \mathbb{E}_{Z \sim \lambda\eta_1 + (1-\lambda)\eta_2}[Z^2] - (\mathbb{E}_{Z \sim \lambda\eta_1 + (1-\lambda)\eta_2}[Z])^2 \\
&= \lambda\mathbb{E}_{Z \sim \eta_1}[Z^2] + (1-\lambda)\mathbb{E}_{Z \sim \eta_2}[Z^2] - (\lambda\mathbb{E}_{Z \sim \eta_1}[Z] + (1-\lambda)\mathbb{E}_{Z \sim \eta_2}[Z])^2 \\
&= \lambda(\mathrm{Var}_{Z \sim \eta_1}[Z] + (\mathbb{E}_{Z \sim \eta_1}[Z])^2) + (1-\lambda)(\mathrm{Var}_{Z \sim \eta_2}[Z] + (\mathbb{E}_{Z \sim \eta_2}[Z])^2) \\
&\quad - (\lambda\mathbb{E}_{Z \sim \eta_1}[Z] + (1-\lambda)\mathbb{E}_{Z \sim \eta_2}[Z])^2.
\end{aligned}$$

∎

This means that to determine whether it is mixture-consistent or not, we should check it on a per-sketch basis, rather than on a per-statistical functional basis.

**Example 4.** Maximum and minimum functional are both mixture-consistent.

*Proof.*

$$\max_{Z \sim \lambda\eta_1 + (1-\lambda)\eta_2} [Z] = \max(\max_{Z \sim \eta_1}[Z], \max_{Z \sim \eta_2}[Z])$$

and

$$\min_{Z \sim \lambda\eta_1 + (1-\lambda)\eta_2} [Z] = \min(\min_{Z \sim \eta_1}[Z], \min_{Z \sim \eta_2}[Z])$$

∎

Since maximum and minimum functionals are mixture consistent, we can construct a nonlinear sketch bellman operator like the one in section C.2. This is possible because there is a nonlinear function $h_\psi$ that ensures the sketch is closed under mixture.

Before demonstrating that a quantile sketch cannot be mixture consistent under any additional sketch, we will first illustrate with the example of a median functional that is not mixture consistent.

**Example 5.** Median sketch is not mixture-consistent.

*Proof.* Let $\lambda = \frac{1}{2}$ and $Z, Y$ be the random variables where $Z \sim 0.2\delta_0 + 0.8\delta_1$ and $Y \sim 0.6\delta_0 + 0.4\delta_k$ for some $0 < k < 1$. Then $\psi_{\mathrm{med}}(Z) = 1$ and $\psi_{\mathrm{med}}(Y) = 0$. However,

$$\mathrm{med}_{X=\frac{Z+Y}{2}}[X] = \psi_{\mathrm{med}}(0.4\delta_0 + 0.2\delta_k + 0.4\delta_1) = k$$

which is dependent in $k$. ∎

**Lemma C.2.** *Quantile sketch cannot be mixture-consistent, under any additional sketch.*

*Proof.* For a given integer $N > 0$ and a quantile level $\alpha \in [0, 1]$, let $\lambda = \frac{1}{2}$ and a random variable $Y \sim p_{y_0}\delta_0 + p_{y_1}\delta_{y_1} + \cdots + p_{y_N}\delta_{y_N} (0 < y_1 < \cdots < y_N < 1)$ where $p_{y_0} > \alpha$ so that $\psi_{q_\alpha}(Y) = 0$. Consider another random variable $Z \sim p_{z_0}\delta_0 + p_{z_1}\delta_1$ where $p_{z_0} < \alpha$ so that $\psi_{q_\alpha}(Z) = 1$. Then the $\alpha-$quantile of the mixture $X = \frac{Y+Z}{2}$ is

$$\psi_{q_\alpha}[X] = y_n \text{ where } n = \min\left\{n \leq N \,\Big|\, \frac{1}{2}\sum_{n'=0}^{n} p_{y_{n'}} + \frac{1}{2}p_{z_0} > \alpha\right\}.$$

Letting $p_{z_0} = 2\alpha - \sum_{n'=0}^{n} p_{y_{n'}}$, we can manipulate $\psi_{q_\alpha}[X]$ to be any value of $y_n$. Hence, $\psi(q_\alpha)[X]$ is a function of all possible outcomes of $Y$.

If there exists a finite number of statistical functionals which make quantile sketch mixture-consistent, then such sketch would uniquely determine the distribution for any $N$. This results in a contradiction that infinite-dimensional distribution space can be represented by a finite number of statistical functional. ∎

**Lemma C.3.** *If a sketch $\psi$ is Bellman closed, then it is mixture-consistent.*

*Proof.* Consider an MDP where initial state $s_0$ has no reward and transits to two state $s_1, s_2$ with probability $\lambda, 1-\lambda$ and reward distribution $\bar{\eta}_1, \bar{\eta}_2$. Since $\psi$ is Bellman closed, $\psi(\bar{\eta}(s_0))$ is a function of $\psi(\bar{\eta}(s_1))$ and $\psi(\bar{\eta}(s_2))$, (i.e., $\psi(\bar{\eta}(s_0)) = g_\psi(\psi(\bar{\eta}(s_1)), \psi(\bar{\eta}(s_2)))$ for some $g_\psi$). Since $\psi(\bar{\eta}(s_0)) = \psi(\lambda\bar{\eta}(s_1) + (1-\lambda)\bar{\eta}(s_2))$, it implies that $\psi$ is mixture-consistent. ∎

Combining the results of Lemma C.2 and Lemma C.3, we prove that a quantile sketch cannot be Bellman closed, no matter what additional sketches are provided.

## D   MAIN PROOF

**Theorem** (4.3). *Quantile functional cannot be Bellman closed under any additional sketch.*

*Proof.*  See Lemma C.2 and Lemma C.3.  ∎

**Lemma** (4.5). *Let $F_{\bar{\eta}}$ be a CDF of the probability distribution $\bar{\eta} \in \mathscr{P}(\mathbb{R})^{\mathcal{S}}$. Then a sketch is Bellman unbiased if and only if the sketch is a homogeneous of degree $k$, i.e., there exists some vector-valued function $h = h(x_1, \cdots, x_k) : \mathcal{X}^k \to \mathbb{R}^N$ such that*

$$\psi(\bar{\eta}) = \int \cdots \int h(x_1, \cdots, x_k) dF_{\bar{\eta}}(x_1) \cdots dF_{\bar{\eta}}(x_k).$$

*Proof.* ($\Rightarrow$) Consider an two-stage MDP with a single action $a$, and an initial state $s_0$ which transits to one of terminal state $\{s_1, \cdots, s_K\}$ with transition kernel $\mathbb{P}(\cdot|s_0, a)$. Assume that the reward $r(s_0) = 0$. Then $\bar{\eta}(s_0) = \sum_{k=1}^K \mathbb{P}(s_k)\delta_{r(s_k)}$. Note that $s'_1, \cdots, s'_k$ are independent and identically distributed random variable in distribution $\mathbb{P}(\cdot|s, a)$.

$$\mathbb{E}_{s' \sim \mathbb{P}(\cdot|s_0, a)} \left[ \phi_\psi \left( \psi\left((\mathcal{B}_r)_\# \bar{\eta}(s'_1)\right), \cdots, \psi\left((\mathcal{B}_r)_\# \bar{\eta}(s'_k)\right) \right) \right] = \psi_{1:N} \left( (\mathcal{B}_r)_\# \mathbb{E}_{s' \sim \mathbb{P}(\cdot|s_0, a)}[\bar{\eta}(s')] \right)$$

$$\implies \mathbb{E}_{s' \sim \mathbb{P}(\cdot|s_0, a)} \left[ \phi_\psi \left( \psi\left(\delta_{r(s'_1)}\right), \cdots, \psi\left(\delta_{r(s'_k)}\right) \right) \right] = \psi \left( \mathbb{E}_{s' \sim \mathbb{P}(\cdot|s_0, a)}[\delta_{r(s')}] \right)$$

$$\implies \mathbb{E}_{s' \sim \mathbb{P}(\cdot|s_0, a)} \left[ \phi_\psi \left( g(s'_1), \cdots, g(s'_k) \right) \right] = \psi \left( \bar{\eta}(s_0) \right)$$

$$\implies \int \cdots \int h(s'_1, \cdots, s'_k) dF_{\bar{\eta}}(s'_1) \cdots dF_{\bar{\eta}}(s'_k) = \psi \left( \bar{\eta}(s_0) \right).$$

($\Leftarrow$)

$$\psi \left( (\mathcal{B}_r)_\# \mathbb{E}_{s' \sim \mathbb{P}(\cdot|s, a)}[\bar{\eta}(s')] \right)$$

$$= \int \cdots \int h(x_1, \cdots, x_k) dF_{(\mathcal{B}_r)_\# \mathbb{E}_{s' \sim \mathbb{P}(\cdot|s, a)}[\bar{\eta}(s')]}(x_1), \cdots, dF_{(\mathcal{B}_r)_\# \mathbb{E}_{s' \sim \mathbb{P}(\cdot|s, a)}[\bar{\eta}(s')]}(x_k)$$

$$= \int \cdots \int h(x_1 + r, \cdots, x_k + r) d \left( \mathbb{E}_{s' \sim \mathbb{P}(\cdot|s, a)} F_{\bar{\eta}(s')}(x_1) \right), \cdots, d \left( \mathbb{E}_{s' \sim \mathbb{P}(\cdot|s, a)} F_{\bar{\eta}(s')}(x_k) \right)$$

$$= \mathbb{E}_{s' \sim \mathbb{P}(\cdot|s, a)} \left[ \int \cdots \int h(x_1 + r, \cdots, x_k + r) dF_{\bar{\eta}(s')}(x_1) \cdots dF_{\bar{\eta}(s')}(x_k) \right]$$

$$= \mathbb{E}_{s' \sim \mathbb{P}(\cdot|s, a)} \left[ \psi \left( (\mathcal{B}_r)_\# [\bar{\eta}(s')] \right) \right]$$

∎

**Theorem** (4.6). *The only finite statistical functionals that are Bellman unbiased and closed are given by the collections of $\psi_1, \ldots, \psi_N$ where its linear span $\{\sum_{n=0}^N \alpha_n \psi_n | \alpha_n \in \mathbb{R}, \forall N\}$ is equal to the set of exponential polynomial functionals $\{\eta \to \mathbb{E}_{Z \sim \eta}[Z^l \exp(\lambda Z)] | l = 0, 1, \ldots, L, \lambda \in \mathbb{R}\}$, where $\psi_0$ is the constant functional equal to $1$. In discount setting, it is equal to the linear span of the set of moment functionals $\{\eta \to \mathbb{E}_{Z \sim \eta}[Z^l] | l = 0, 1, \ldots, L\}$ for some $L \leq N$.*

*Proof.*  Our proof is mainly based on the proof techniques of Rowland et al. (2019) and we describe in an extended form. Since their proof also considers the discounted setting, we will define $\mathcal{B}_{r, \gamma}(x) = r + \gamma x$ for discount factor $\gamma \in [0, 1)$. By assumption of Bellman closedness, $\psi_n \left( (\mathcal{B}_{r, \gamma})_\# \bar{\eta}(s') \right)$ will be written as

$g(r, \gamma, \psi_{1:N}(\bar{\eta}(s')))$ for some $g$. By assumption of Bellman unbiasedness and Lemma 4.5, both $\psi_{1:N}(\bar{\eta}(s'))$ and $\psi_n\left((\mathcal{B}_{r,\gamma})_{\#}\bar{\eta}(s')\right)$ are affine as functions of the distribution $\bar{\eta}(s')$,

$$
\begin{aligned}
&\psi_{1:N}(\alpha\bar{\eta}_1(s') + (1-\alpha)\bar{\eta}_2(s'))\\
&= \mathbb{E}_{Z_i \sim \alpha\bar{\eta}_1(s') + (1-\alpha)\bar{\eta}_2(s')}[h_{1:N}(\bar{Z}_1, \cdots, \bar{Z}_k)]\\
&= \alpha\mathbb{E}_{\bar{Z}_i \sim \bar{\eta}_1(s')}[h_{1:N}(\bar{Z}_1, \cdots, \bar{Z}_k)] + (1-\alpha)\mathbb{E}_{\bar{Z}_i \sim \bar{\eta}_2(s')}[h_{1:N}(\bar{Z}_1, \cdots, \bar{Z}_k)]\\
&= \alpha\psi_{1:N}(\bar{\eta}_1(s')) + (1-\alpha)\psi_{1:N}(\bar{\eta}_2(s'))
\end{aligned}
$$

and

$$
\begin{aligned}
&\psi_n\left((\mathcal{B}_{r,\gamma})_{\#}(\alpha\bar{\eta}_1(s') + (1-\alpha)\bar{\eta}_2(s'))\right)\\
&= \mathbb{E}_{Z_i \sim \alpha\bar{\eta}_1(s') + (1-\alpha)\bar{\eta}_2(s')}[h_n(r + \gamma\bar{Z}_1, \cdots, r + \gamma\bar{Z}_k)]\\
&= \alpha\mathbb{E}_{\bar{Z}_i \sim \bar{\eta}_1(s')}[h_n(r + \gamma\bar{Z}_1, \cdots, r + \gamma\bar{Z}_k)] + (1-\alpha)\mathbb{E}_{\bar{Z}_i \sim \bar{\eta}_2(s')}[h_n(r + \gamma\bar{Z}_1, \cdots, r + \gamma\bar{Z}_k)]\\
&= \alpha\psi_n\left((\mathcal{B}_{r,\gamma})_{\#}\bar{\eta}_1(s')\right) + (1-\alpha)\psi_n\left((\mathcal{B}_{r,\gamma})_{\#}\bar{\eta}_2(s')\right)
\end{aligned}
$$

Therefore, $g(r, \gamma, \cdot)$ is also affine on the convex codomain of $\psi_{1:N}$. Thus, we have

$$
\mathbb{E}_{\bar{Z}_i \sim \bar{\eta}}[\phi_{\psi_n}(r + \gamma\bar{Z}_1, \cdots, r + \gamma\bar{Z}_k)] = a_0(r, \gamma) + \sum_{n'=1}^{N} a_{n'}(r, \gamma)\mathbb{E}_{\bar{Z}_i \sim \bar{\eta}}[\phi_{\psi_{n'}}(\bar{Z}_1, \cdots, \bar{Z}_k)]
$$

for some function $a_{0:N} : \mathbb{R} \times [0, 1] \to \mathbb{R}$. By taking $\bar{\eta}(s') = \delta_x$, we obtain

$$
\phi_{\psi_n}(r + \gamma x, \cdots, r + \gamma x) = a_0(r, \gamma) + \sum_{n'=1}^{N} a_{n'}(r, \gamma)\phi_{\psi_{n'}}(x, \cdots, x).
$$

According to Engert (1970), for any translation invariant finite-dimensional space is spanned by a set of function of the form

$$
\{x \mapsto x^l \exp(\lambda_j x) | \; j \in [J], 0 \leq l \leq L\}
$$

for some finite subset $\{\lambda_1, \cdots, \lambda_J\}$ of $\mathbb{C}$. Hence, each function $x \mapsto \phi_{\psi_n}(x, \cdots, x)$ is expressed as linear combination of exponential polynomial functions. In addition, the linear combination of $\phi_{\psi_n}$ should be closed under composition with for any discount factor $\gamma \in [0, 1]$, all $\lambda_j$ should be zero. Hence, the linear combination of $\phi_{\psi_1}, \cdots, \phi_{\psi_N}$ must be equal to the span of $\{x \mapsto x^l | \; 0 \leq l \leq L\}$ for some $L \in \mathbb{N}$.

∎

**Lemma** (6.3). *Consider a fixed $k \in [K]$ and a fixed $h \in [H]$. Let $\mathcal{Z}_h^k = \{(s_h^\tau, a_h^\tau)\}_{\tau \in [k-1]}$ and $\mathcal{D}_{h,\bar{\eta}}^k = \left\{\left(s_h^\tau, a_h^\tau, \psi_{1:N}\left((\mathcal{B}_{r_{h'}}^\tau)_{\#}\bar{\eta}(s_{h'+1}^\tau)\right)\right)\right\}_{\tau \in [k-1]}$ for any $\bar{\eta} : \mathcal{S} \to \mathscr{P}([0, H])$. Define $\tilde{f}_{h,\bar{\eta}}^k = \arg\min_{f \in \mathcal{F}^N} \|f\|_{\mathcal{D}_{h,\bar{\eta}}^k}^2$. For any $\bar{\eta}$ and $\delta \in (0, 1)$, there is an event $\mathcal{E}(\bar{\eta}, \delta)$ such that conditioned on $\mathcal{E}(\bar{\eta}, \delta)$, with probability at least $1 - \delta$, for any $\bar{\eta}' : \mathcal{S} \to \mathscr{P}([0, H])$ with $\|\psi_{1:N}(\bar{\eta}') - \psi_{1:N}(\bar{\eta})\|_{\infty,1} \leq 1/T$ or $\sum_{n=1}^{N} \|\psi_n(\bar{\eta}') - \psi_n(\bar{\eta})\|_{\infty} \leq 1/T$, we have*

$$
\left\|\tilde{f}_{h,\bar{\eta}'}(\cdot, \cdot) - \psi_{1:N}\left((\mathcal{B}_{r(\cdot,\cdot)})_{\#}[\mathbb{P}\bar{\eta}'](\cdot, \cdot)\right)\right\|_{\mathcal{Z}_h^k} \leq c'\left(N^{\frac{1}{2}}H\sqrt{\log(1/\delta) + \log\mathcal{N}(\mathcal{F}^N, 1/T)}\right)
$$

*for some constant $c' > 0$.*

*Proof.* Define the sketch of target $f_{\bar{\eta}} : \mathcal{S} \times \mathcal{A} \to \mathbb{R}^N$,

$$
f_{\bar{\eta}}(\cdot, \cdot) := \psi_{1:N}\left((\mathcal{B}_{r(\cdot,\cdot)})_{\#}[\mathbb{P}\bar{\eta}](\cdot, \cdot)\right)
$$

for all $i \in [N]$.

For any $f \in \mathcal{F}$,

$$\|f\|^2_{\mathcal{D}^k_{h,\bar{\eta}'}} - \|f_{\bar{\eta}'}\|^2_{\mathcal{D}^k_{h,\bar{\eta}'}}$$

$$= \sum_{n=1}^N \sum_{s^\tau_h, a^\tau_h \in \mathcal{Z}^k_{h,\bar{\eta}'}} \left( f^{(n)}(s^\tau_h, a^\tau_h) - \psi_n\left((\mathcal{B}_{r^\tau_h})_{\#}\bar{\eta}'(s^\tau_{h+1})\right) \right)^2 - \left( f^{(n)}_{\bar{\eta}'}(s^\tau_h, a^\tau_h) - \psi_n\left((\mathcal{B}_{r^\tau_h})_{\#}\bar{\eta}'(s^\tau_{h+1})\right) \right)^2$$

$$= \sum_{n=1}^N \sum_{s^\tau_h, a^\tau_h \in \mathcal{Z}^k_{h,\bar{\eta}'}} (f^{(n)}(s^\tau_h, a^\tau_h) - f^{(n)}_{\bar{\eta}'}(s^\tau_h, a^\tau_h))^2$$

$$+ 2(f^{(n)}(s^\tau_h, a^\tau_h) - f^{(n)}_{\bar{\eta}'}(s^\tau_h, a^\tau_h))\left( f^{(n)}_{\bar{\eta}'}(s^\tau_h, a^\tau_h) - \psi_n\left((\mathcal{B}_{r^\tau_h})_{\#}\bar{\eta}'(s^\tau_{h+1})\right) \right)$$

$$\geq \|f - f_{\bar{\eta}'}\|^2_{\mathcal{Z}^k_h} - 4\sum_{n=1}^N \|f^{(n)}_{\bar{\eta}} - f^{(n)}_{\bar{\eta}'}\|_\infty (H+1)|\mathcal{Z}^k_h|$$

$$+ \sum_{n=1}^N \sum_{s^\tau_h, a^\tau_h \in \mathcal{Z}^k_{h,\bar{\eta}'}} \bigg[ \underbrace{2(f^{(n)}(s^\tau_h, a^\tau_h) - f^{(n)}_{\bar{\eta}}(s^\tau_h, a^\tau_h))\left( f^{(n)}_{\bar{\eta}}(s^\tau_h, a^\tau_h) - \psi_n\left((\mathcal{B}_{r^\tau_h})_{\#}\bar{\eta}(s^\tau_{h+1})\right) \right)}_{\chi^\tau_h(f^{(n)})} \bigg]$$

$$\geq \|f - f_{\bar{\eta}'}\|^2_{\mathcal{Z}^k_h} - 4N(H+1) - \bigg| \sum_{n=1}^N \sum_{s^\tau_h, a^\tau_h \in \mathcal{Z}^k_{h,\bar{\eta}'}} \chi^\tau_h(f^{(n)}) \bigg|.$$

For the first inequality, we change the second term from $\bar{\eta}'$ to $\bar{\eta}$ which are the $\epsilon$-covers. Notice that $AC - BC' \geq -|AC - BC'| \geq -|(A-B)C| - |(A-B)C'| \geq -2|A-B||\max(C, C')|$.

$$(f^{(n)}(s^\tau_h, a^\tau_h) - f^{(n)}_{\bar{\eta}'}(s^\tau_h, a^\tau_h))\left( f^{(n)}_{\bar{\eta}'}(s^\tau_h, a^\tau_h) - \psi_n\left((\mathcal{B}_{r^\tau_h})_{\#}\bar{\eta}'(s^\tau_{h+1})\right) \right)$$

$$- (f^{(n)}(s^\tau_h, a^\tau_h) - f^{(n)}_{\bar{\eta}}(s^\tau_h, a^\tau_h))\left( f^{(n)}_{\bar{\eta}}(s^\tau_h, a^\tau_h) - \psi_n\left((\mathcal{B}_{r^\tau_h})_{\#}\bar{\eta}(s^\tau_{h+1})\right) \right)$$

$$\geq -2\|f^{(n)}_{\bar{\eta}'}(s^\tau_h, a^\tau_h) - f^{(n)}_{\bar{\eta}}(s^\tau_h, a^\tau_h)\|$$

$$\times \max\left( \left| f^{(n)}_{\bar{\eta}'}(s^\tau_h, a^\tau_h) - \psi_n\left((\mathcal{B}_{r^\tau_h})_{\#}\bar{\eta}'(s^\tau_{h+1})\right) \right|, \left| f^{(n)}_{\bar{\eta}}(s^\tau_h, a^\tau_h) - \psi_n\left((\mathcal{B}_{r^\tau_h})_{\#}\bar{\eta}(s^\tau_{h+1})\right) \right| \right)$$

$$\geq -2\|f^{(n)}_{\bar{\eta}'}(s^\tau_h, a^\tau_h) - f^{(n)}_{\bar{\eta}}(s^\tau_h, a^\tau_h)\|(H+1)$$

For the second inequality, consider $\bar{\eta}' : \mathcal{S} \to \mathscr{P}([0, H])$ with $\sum_{n=1}^N \|\psi_n(\bar{\eta}') - \psi_n(\bar{\eta})\|_\infty \leq 1/T$. We have

$$\|f^{(n)}_{\bar{\eta}} - f^{(n)}_{\bar{\eta}'}\|_\infty = \max_{s,a} \bigg| \sum_{n'=1}^n H^{n'}[\psi_{n'}([\mathbb{P}\bar{\eta}](s,a)) - \psi_{n'}([\mathbb{P}\bar{\eta}'](s,a))]r^{n-n'}/H^{n-1} \bigg|$$

$$\leq \sum_{n'=1}^n \max_{s'} \left| \psi_{n'}(\bar{\eta}(s')) - \psi_{n'}(\bar{\eta}'(s')) \right|$$

$$\leq 1/T.$$

Defining $\mathbb{F}_h^k$ as the filtration induced by the sequence $\{(s_{h'}^\tau, a_{h'}^\tau)\}_{\tau, h' \in [k-1] \times [H]} \cup \{(s_1^k, a_1^k), (s_2^k, a_2^k), \ldots, (s_h^k, a_h^k)\}$, notice that

$$\mathbb{E}\Big[\sum_{n=1}^N \chi_h^\tau(f^{(n)})\Big|\mathbb{F}_h^\tau\Big]$$

$$= \sum_{n=1}^N 2(f^{(n)}(s_h^\tau, a_h^\tau) - f_{\bar\eta}^{(n)}(s_h^\tau, a_h^\tau))(f_{\bar\eta}^{(n)}(s_h^\tau, a_h^\tau) - \mathbb{E}\Big[\psi_n\big((\mathcal{B}_{r_h^\tau})_\# \bar\eta(s_{h+1}^\tau)\big)\Big|\mathbb{F}_h^\tau\Big])$$

$$= \sum_{n=1}^N 2(f^{(n)}(s_h^\tau, a_h^\tau) - f_{\bar\eta}^{(n)}(s_h^\tau, a_h^\tau))(f_{\bar\eta}^{(n)}(s_h^\tau, a_h^\tau) - \mathbb{E}_{s_{h+1}^\tau \sim \mathbb{P}_h(\cdot|s_h^\tau, a_h^\tau)}\Big[\psi_n\big((\mathcal{B}_{r_h^\tau})_\# \bar\eta(s_{h+1}^\tau)\big)\Big])$$

$$= \sum_{n=1}^N 2(f^{(n)}(s_h^\tau, a_h^\tau) - f_{\bar\eta}^{(n)}(s_h^\tau, a_h^\tau))(f_{\bar\eta}^{(n)}(s_h^\tau, a_h^\tau) - \psi_n\big((\mathcal{B}_{r_h^\tau})_\# \mathbb{E}_{s_{h+1}^\tau \sim \mathbb{P}_h(\cdot|s_h^\tau, a_h^\tau)}[\bar\eta(s_{h+1}^\tau)]\big))$$

$$= 0$$

and

$$\Big|\sum_{n=1}^N \chi_h^\tau(f^{(n)})\Big| = \Big|\sum_{n=1}^N 2(f^{(n)}(s_h^\tau, a_h^\tau) - f_{\bar\eta}^{(n)}(s_h^\tau, a_h^\tau))(f_{\bar\eta}^{(n)}(s_h^\tau, a_h^\tau) - \psi_n\big((\mathcal{B}_{r_h^\tau})_\# \bar\eta(s_{h+1}^\tau)\big))\Big|$$

$$\leq \max_{n \in [N]} \Big\{2(f_{\bar\eta}^{(n)}(s_h^\tau, a_h^\tau) - \psi_n\big((\mathcal{B}_{r_h^\tau})_\# \bar\eta(s_{h+1}^\tau)\big))\Big\} \sum_{n=1}^N \Big|f^{(n)}(s_h^\tau, a_h^\tau) - f_{\bar\eta}^{(n)}(s_h^\tau, a_h^\tau)\Big|$$

$$\leq 2(H+1) \sum_{n=1}^N \Big|f^{(n)}(s_h^\tau, a_h^\tau) - f_{\bar\eta}^{(n)}(s_h^\tau, a_h^\tau)\Big|$$

In third equality, we emphasize that only Bellman unbiased sketch can derive the martingale difference sequence which induce the concentration result. Since every moment functional is commutable with mixing operation, the transformation $\phi_{\psi_n}$ in Definition 4.4 is identity for all $n \in [N]$. Hence, we choose the sketch as moment which already knows $\phi_\psi$.

By Azuma-Hoeffding inequality,

$$\mathbb{P}\Big[\Big|\sum_{(\tau,h) \in [k-1] \times [H]} \sum_{n=1}^N \chi_h^\tau(f^{(n)})\Big| \geq \epsilon\Big] \leq 2\exp\Big(-\frac{\epsilon^2}{2(2(H+1))^2 \sum_{(\tau,h) \in [k-1] \times [H]} \big(\sum_{n=1}^N |f^{(n)} - f_{\bar\eta}^{(n)}|\big)^2}\Big)$$

$$\leq 2\exp\Big(-\frac{\epsilon^2}{2(2(H+1))^2 \sum_{(\tau,h) \in [k-1] \times [H]} \big(N \sum_{n=1}^N |f^{(n)} - f_{\bar\eta}^{(n)}|^2\big)}\Big)$$

$$= 2\exp\Big(-\frac{\epsilon^2}{2N(2(H+1))^2 \|f - f_{\bar\eta}\|_{\mathcal{Z}_h^k}^2}\Big)$$

where the second inequality follows from the Cauchy-Schwartz inequality.

We set

$$\epsilon = \sqrt{8N(H+1)^2 \|f - f_{\bar\eta}\|_{\mathcal{Z}_h^k}^2 \log\Big(\frac{\mathcal{N}(\mathcal{F}^N, 1/T)}{\delta}\Big)}$$

With union bound for all $f \in \mathcal{C}(\mathcal{F}^N, 1/T)$, with probability at least $1 - \delta$,

$$\Big|\sum_{(\tau,h) \in [k-1] \times [H]} \sum_{n=1}^N \chi_h^\tau(f^{(n)})\Big| \leq c' N^{\frac{1}{2}}(H+1) \|f - f_{\bar\eta}\|_{\mathcal{Z}_h^k} \sqrt{\log\Big(\frac{\mathcal{N}(\mathcal{F}^N, 1/T)}{\delta}\Big)}$$

for some constant $c' > 0$.

For all $f \in \mathcal{F}^N$, there exists $g \in \mathcal{C}(\mathcal{F}^N, 1/T)$, such that $\|f-g\|_{\infty,1} \leq 1/T$ or $\sum_{n=1}^N \|f^{(n)} - g^{(n)}\|_\infty \leq 1/T$ for all $n \in [N]$,

$$\left| \sum_{(\tau,h) \in [k-1] \times [H]} \sum_{n=1}^N \chi_h^\tau(f^{(n)}) \right| \leq \left| \sum_{(\tau,h) \in [k-1] \times [H]} \sum_{n=1}^N \chi_h^\tau(g^{(n)}) \right| + 2(H+1)|\mathcal{Z}_h^k| \sum_{n=1}^N \frac{1}{T}$$

$$\leq c' N^{\frac{1}{2}}(H+1)\|g - f_{\bar\eta}\|_{\mathcal{Z}_h^k} \sqrt{\log\left(\frac{\mathcal{N}(\mathcal{F}^N, 1/T)}{\delta}\right)} + 2N(H+1)$$

$$\leq c' N^{\frac{1}{2}}(H+1)(\|f - f_{\bar\eta}\|_{\mathcal{Z}_h^k} + 1)\sqrt{\log\left(\frac{\mathcal{N}(\mathcal{F}^N, 1/T)}{\delta}\right)} + 2N(H+1)$$

$$\leq c' N^{\frac{1}{2}}(H+1)(\|f - f_{\bar\eta'}\|_{\mathcal{Z}_h^k} + 2)\sqrt{\log\left(\frac{\mathcal{N}(\mathcal{F}^N, 1/T)}{\delta}\right)} + 2N(H+1)$$

where the third inequality follows from,

$$\|f - g\|_{\mathcal{Z}_h^k}^2 \leq \sum_{n=1}^N \sum_{(\tau,h) \in [k-1] \times [H]} |f^{(n)}(s_h^\tau, a_h^\tau) - g^{(n)}(s_h^\tau, a_h^\tau)|^2$$

$$\leq NT\left(\frac{1}{T}\right)^2$$

$$\leq 1.$$

Recall that $\tilde{f}_{h,\eta'}^k = \arg\min_{f \in \mathcal{F}} \|f\|_{\mathcal{D}_{h,\eta'}^k}^2$. We have $\|\tilde{f}_{h,\eta'}^k\|_{\mathcal{D}_{h,\eta'}^k}^2 - \|f_{\bar\eta'}\|_{\mathcal{D}_{h,\eta'}^k}^2 \leq 0$, which implies,

$$0 \geq \|\tilde{f}_{h,\bar\eta'}^k\|_{\mathcal{D}_{h,\bar\eta'}^k}^2 - \|f_{\bar\eta'}\|_{\mathcal{D}_{h,\bar\eta'}^k}^2$$

$$= \|\tilde{f}_{h,\bar\eta'}^k - f_{\bar\eta'}\|_{\mathcal{Z}_h^k}^2$$

$$+ 2\sum_{n=1}^N \sum_{(\tau,h) \in [k-1] \times [H]} \left[ ((\tilde{f}_{h,\bar\eta'}^k)^{(n)}(s_h^\tau, a_h^\tau) - f_{\bar\eta'}^{(n)}(s_h^\tau, a_h^\tau))(f_{\bar\eta'}^{(n)}(s_h^\tau, a_h^\tau) - \psi_n\left((\mathcal{B}_{r_h^\tau})_\# \bar\eta'(s_{h+1}^\tau)\right)) \right]$$

$$\geq \|\tilde{f}_{h,\bar\eta'}^k - f_{\bar\eta'}\|_{\mathcal{Z}_h^k}^2 - c' N^{\frac{1}{2}}(H+1)(\|\hat{f}_{h,\bar\eta'}^k - f_{\bar\eta'}\|_{\mathcal{Z}_h^k} + 2)\sqrt{\log(2/\delta) + \log\mathcal{N}(\mathcal{F}^N, 1/T)} - 6N(H+1).$$

Recall that if $x^2 - 2ax - b \leq 0$ holds for constant $a, b > 0$, then $x \leq a + \sqrt{a^2 + b} \leq c' \cdot a$ for some constant $c' > 0$.

Hence,

$$\|\tilde{f}_{h,\eta'}^k - f_{\bar\eta'}\|_{\mathcal{Z}_h^k} \leq c'(N^{\frac{1}{2}}H\sqrt{\log(1/\delta) + \log\mathcal{N}(\mathcal{F}^N, 1/T)})$$

for some constant $c' > 0$. ∎

**Lemma** (6.4). *Let* $(\mathcal{F}^N)_h^k = \{f \in \mathcal{F}^N | \|f - \tilde{f}_{h,\bar\eta}^k\|_{\mathcal{Z}_h^k}^2 \leq \beta(\mathcal{F}^N, \delta)\}$, *where*

$$\beta(\mathcal{F}^N, \delta) \geq c' \cdot NH^2(\log(T/\delta) + \log\mathcal{N}(\mathcal{F}^N, 1/T))$$

*for some constant* $c' > 0$. *Then with probability at least* $1 - \delta/2$, *for all* $k, h \in [K] \times [H]$, *we have*

$$\psi_n\left((\mathcal{B}_{r_h(\cdot,\cdot)})_\# [\mathbb{P}_h \bar\eta_{h+1}^k](\cdot,\cdot)\right) \in (\mathcal{F}^N)_h^k$$

*Proof.* For all $(k, h) \in [K] \times [H]$,

$$\mathbf{S} := \begin{cases} \left\{ \left( \min\{f^{(1)}(\cdot,\cdot) + b_{h+1}^k(\cdot,\cdot), H\} \right) \Big| f \in \mathcal{C}(\mathcal{F}^N, 1/T) \right\} \cup \{0\} & n = 1 \\ \left\{ \left( \min\{f^{(n)}(\cdot,\cdot), H\} \right) \Big| f \in \mathcal{C}(\mathcal{F}^N, 1/T) \right\} \cup \{0\} & 2 \leq n \leq N \end{cases}$$

is a $(1/T)$-cover of $\psi_{1:N}(\eta_{h+1}^k(\cdot, \cdot))$ where

$$\psi_{1:N}(\eta_{h+1}^k(\cdot, \cdot)) = \begin{cases} \min\{(f_{h+1}^k)^{(1)}(\cdot, \cdot) + b_{h+1}^k(\cdot, \cdot), H\} & n = 1 \text{ and } h < H \\ \min\{(f_{h+1}^k)^{(n)}(\cdot, \cdot), H\} & 2 \le n \le N \text{ and } h < H \\ \mathbf{0}^N & h = H \end{cases},$$

i.e., there exists $\psi_{1:N}(\eta) \in \mathbf{S}$ such that $\|\psi_{1:N}(\eta) - \psi_{1:N}(\eta_{h+1}^k)\|_{\infty,1} \le 1/T$. This implies

$$\bar{\mathbf{S}} := \left\{ \psi_{1:N}\left( \eta(\cdot, \arg\max_{a \in \mathcal{A}} \psi_1(\eta(\cdot, a))) \right) \mid \psi_{1:N}(\eta) \in \mathbf{S} \right\}$$

is a $(1/T)$-cover of $\psi_{1:N}(\bar{\eta}_{h+1}^k)$ with $\log(|\bar{\mathbf{S}}|) \le \log \mathcal{N}(\mathcal{F}^N, 1/T)$.

For each $\psi_{1:N}(\bar{\eta}) \in \bar{\mathbf{S}}$, let $\mathcal{E}(\bar{\eta}, \delta/2|\bar{\mathbf{S}}|T)$ be the event defined in Lemma 6.3. By union bound for all $\psi_{1:N}(\bar{\eta}) \in \bar{\mathbf{S}}$, we have $\Pr[\bigcap_{\psi_{1:N}(\bar{\eta}) \in \bar{\mathbf{S}}} \mathcal{E}(\bar{\eta}, \delta/2|\bar{\mathbf{S}}|T)] \ge 1 - \delta/2T$.

Let $\psi_{1:N}(\bar{\eta}) \in \bar{\mathbf{S}}$ such that $\|\psi_{1:N}(\bar{\eta}) - \psi_{1:N}(\bar{\eta}_{h+1}^k)\|_{\infty,1} \le 1/T$. Conditioned on $\bigcap_{s_N(\bar{\eta}) \in \bar{\mathbf{S}}} \mathcal{E}(\bar{\eta}, \delta/2|\bar{\mathbf{S}}|T)$ and by Lemma 6.3, we have

$$\left\| \tilde{f}_{h,\bar{\eta}}^k(\cdot, \cdot) - \psi_{1:N}\left( (\mathcal{B}_{r_h(\cdot, \cdot)})_\# [\mathbb{P}_h \bar{\eta}_{h+1}^k](\cdot, \cdot) \right) \right\|_{\mathcal{Z}_h^k}^2 \le c' \left( NH^2(\log(T/\delta) + \log \mathcal{N}(\mathcal{F}^N, 1/T)) \right)$$

for some constant $c' > 0$.

By union bound for all $(k, h) \in [K] \times [H]$, we have $\psi_{1:N}\left( (\mathcal{B}_{r_h(\cdot, \cdot)})_\# [\mathbb{P}_h \bar{\eta}_{h+1}^k](\cdot, \cdot) \right) \in (\mathcal{F}^N)_h^k$ with probability $1 - \delta/2$. ∎

**Lemma D.1.** *Let* $Q_h^k(s, a) := \min\{H, \tilde{f}_h^k(s, a) + b_h^k(s, a)\}$ *for some bonus function* $b_h^k(s, a)$ *for all* $(s, a) \in \mathcal{S} \times \mathcal{A}$. *If* $b_h^k(s, a) \ge w^{(1)}((\mathcal{F}^N)_h^k, s, a)$, *then with probability at least* $1 - \delta/2$,

$$Q_h^*(s, a) \le Q_h^k(s, a) \text{ and } V_h^*(s) \le V_h^k(s)$$

*for all* $(k, h) \in [K] \times [H]$, *for all* $(s, a) \in \mathcal{S} \times \mathcal{A}$.

*Proof.* We use induction on $h$ from $h = H$ to 1 to prove the statement. Let $\mathcal{E}$ be the event that for $(k, h) \in [K] \times [H]$, $\psi_{1:N}\left( (\mathcal{B}_{r_h(\cdot, \cdot)})_\# [\mathbb{P}_h \bar{\eta}_{h+1}^k](\cdot, \cdot) \right) \in (\mathcal{F}^N)_h^k$. By Lemma 6.4, $\Pr|\mathcal{E}| \ge 1 - \delta/2$. In the rest of the proof, we condition on $\mathcal{E}$.

When $h = H + 1$, the desired inequality holds as $Q_{H+1}^*(s, a) = V_{H+1}^*(s) = Q_{H+1}^k(s, a) = V_{H+1}^k(s) = 0$. Now, assume that $Q_{h+1}^*(s, a) \le Q_{h+1}^k(s, a)$ and $V_{h+1}^*(s) \le V_{h+1}^k(s)$ for some $h \in [H]$. Then, for all $(s, a) \in \mathcal{S} \times \mathcal{A}$,

$$\begin{aligned} Q_h^*(s, a) &= \min\{H, r_h(s, a) + [\mathbb{P}_h V_{h+1}^*](s, a)\} \\ &\le \min\{H, r_h(s, a) + [\mathbb{P}_h V_{h+1}^k](s, a)\} \\ &\le \min\{H, \tilde{f}_h^k(s, a) + w^{(1)}(\mathcal{F}_h^k, s, a)\} \\ &= \min\{H, Q_h^k(s, a) - b_h^k(s, a) + w^{(1)}(\mathcal{F}_h^k, s, a)\} \\ &\le Q_h^k(s, a) \end{aligned}$$

∎

**Lemma D.2** (Regret decomposition)**.** *With probability at least* $1 - \delta/4$, *we have*

$$Reg(K) \le \sum_{k=1}^K \sum_{h=1}^H (2b_h^k(s_h^k, a_h^k) + \xi_h^k),$$

*where* $\xi_h^k = [\mathbb{P}_h(V_{h+1}^k - V_{h+1}^{\pi^k})](s_h^k, a_h^k) - (V_{h+1}^k(s_{h+1}^k) - V_{h+1}^{\pi^k}(s_{h+1}^k))$ *is a martingale difference sequence with respect to the filtration* $\mathbb{F}_h^k$ *induced by the history* $\mathbb{H}_h^k$.

*Proof.* We condition on the above event $\mathcal{E}$ in the rest of the proof. For all $(k, h) \in [K] \times [H]$, we have

$$\left\| \tilde{f}^k_{h,\bar{\eta}}(\cdot, \cdot) - \psi_{1:N}\left( (\mathcal{B}_{r_h(\cdot,\cdot)})_\# [\mathbb{P}_h \bar{\eta}^k_{h+1}](\cdot, \cdot) \right) \right\|^2_{\mathcal{Z}^k_h} \leq \beta(\mathcal{F}^N, \delta).$$

Recall that $(\mathcal{F}^N)^k_h = \{ f \in \mathcal{F}^N \mid \| f - \tilde{f}^k_{h,\bar{\eta}} \|^2_{\mathcal{Z}^k_h} \leq \beta(\mathcal{F}^N, \delta) \}$ is the confidence region. Since $\psi_{1:N}\left( (\mathcal{B}_{r_h(\cdot,\cdot)})_\# [\mathbb{P}_h \bar{\eta}^k_{h+1}](\cdot, \cdot) \right) \in (\mathcal{F}^N)^k_h$, then by the definition of width function $w^{(1)}(\mathcal{F}^k_h, s, a)$, for $(k, h) \in [K] \times [H]$, we have

$$w^{(1)}(\mathcal{F}^k_h, s, a) \geq \left| \psi_1\left( (\mathcal{B}_{r_h(s,a)})_\# [\mathbb{P}_h \bar{\eta}^k_{h+1}](s, a) \right) - (\tilde{f}^k_{h,\bar{\eta}})^{(1)}(s, a) \right|$$
$$= \left| r_h(s, a) + [\mathbb{P}_h V^k_{h+1}](s, a) - (\tilde{f}^k_{h,\bar{\eta}})^{(1)}(s, a) \right|.$$

Recall that $Q^*_h(\cdot, \cdot) \leq Q^k_h(\cdot, \cdot)$.

$$\text{Reg}(K) = \sum_{k=1}^K V^\star_1(s^k_1) - V^{\pi^k}_1(s^k_1)$$

$$\leq \sum_{k=1}^K V^k_1(s^k_1) - V^{\pi^k}_1(s^k_1)$$

$$= \sum_{k=1}^K Q^k_1(s^k_1, a^k_1) - Q^{\pi^k}_1(s^k_1, a^k_1)$$

$$= \sum_{k=1}^K Q^k_1(s^k_1, a^k_1) - (r_1(s^k_1, a^k_1) + [\mathbb{P}_1 V^k_2](s^k_1, a^k_1)) + (r_1(s^k_1, a^k_1) + [\mathbb{P}_1 V^k_2](s^k_1, a^k_1))$$
$$\quad - Q^{\pi^k}_1(s^k_1, a^k_1)$$

$$\leq \sum_{k=1}^K w^{(1)}((\mathcal{F}^N)^k_1, s^k_1, a^k_1) + b^k_1(s^k_1, a^k_1) + [\mathbb{P}_1(V^k_2 - V^{\pi^k}_2)](s^k_1, a^k_1)$$

$$\leq \sum_{k=1}^K w^{(1)}((\mathcal{F}^N)^k_1, s^k_1, a^k_1) + b^k_1(s^k_1, a^k_1) + (V^k_2(s^k_2) - V^{\pi^k}_2(s^k_2)) + \xi^k_1$$

$$\vdots$$

$$\leq \sum_{k=1}^K \sum_{h=1}^H (w^{(1)}((\mathcal{F}^N)^k_h, s^k_h, a^k_h) + b^k_h(s^k_h, a^k_h) + \xi^k_h)$$

$$\leq \sum_{k=1}^K \sum_{h=1}^H (2 b^k_h(s^k_h, a^k_h) + \xi^k_h)$$

∎

It remains to bound $\sum_{k=1}^K \sum_{h=1}^H b^k_h(s^k_h, a^k_h)$, for which we will exploit fact that $\mathcal{F}^N$ has bounded eluder dimension.

**Lemma D.3.** *If $b^k_h(s, a) \geq w^{(1)}((\mathcal{F}^N)^k_h, s, a)$ for all $(s, a) \in \mathcal{S} \times \mathcal{A}$ and $k \in [K]$ where*

$$(\mathcal{F}^N)^k_h = \{ f \in \mathcal{F}^N \mid \| f - \tilde{f}^k_{h,\bar{\eta}} \|^2_{\mathcal{Z}^k_h} \leq \beta(\mathcal{F}^N, \delta) \},$$

*then*

$$\sum_{k=1}^K \sum_{h=1}^H \mathbf{1}\{ b^k_h(s^k_h, a^k_h) > \epsilon \} \leq \left( \frac{4\beta(\mathcal{F}^N, \delta)}{\epsilon^2} + 1 \right) dim_E(\mathcal{F}^N, \epsilon)$$

*for some constant $c > 0$.*

*Proof.* We first want to show that for any sequence $\{(s_1, a_1), \ldots, (s_\kappa, a_\kappa)\} \subseteq \mathcal{S} \times \mathcal{A}$, there exists $j \in [\kappa]$ such that $(s_j, a_j)$ is $\epsilon$-dependent on at least $L = \lceil (\kappa - 1)/\dim_E(\mathcal{F}^N, \epsilon) \rceil$ disjoint subsequences in $\{(s_1, a_1), \ldots, (s_{j-1}, a_{j-1})\}$ with respect to $\mathcal{F}^N$. We demonstrate this by using the following procedure. Start with $L$ disjoint subsequences of $\{(s_1, a_1), \ldots, (s_{j-1}, a_{j-1})\}$, $\mathcal{B}_1, \mathcal{B}_2, \ldots, \mathcal{B}_L$, which are initially empty. For each $j$, if $(s_j, a_j)$ is $\epsilon$-dependent on every $\mathcal{B}_1, \ldots, \mathcal{B}_L$, we achieve our goal so we stop the process. Else, we choose $i \in [L]$ such that $(s_j, a_j)$ is $\epsilon$-independent on $\mathcal{B}_i$ and update $\mathcal{B}_i \leftarrow \mathcal{B}_i \cup \{(s_j, a_j)\}$, $j \leftarrow j + 1$. Since every element of $\mathcal{B}_i$ is $\epsilon$-independent on its predecessors, $|\mathcal{B}_i|$ cannot get bigger than $\dim_E(\mathcal{F}^N, \epsilon)$ at any point in this process. Therefore, the process stops at most step $j = L\dim_E(\mathcal{F}^N, \epsilon) + 1 \leq \kappa$.

Now we want to show that if for some $j \in [\kappa]$ such that $b_h^k(s_j, a_j) > \epsilon$, then $(s_j, a_j)$ is $\epsilon$-dependent on at most $4\beta(\mathcal{F}^N, \delta)/\epsilon^2$ disjoint subsequences in $\{(s_1, a_1), \ldots, (s_{j-1}, a_{j-1})\}$ with respect to $\mathcal{F}^N$. If $b_h^k(s_j, a_j) > \epsilon$ and $(s_j, a_j)$ is $\epsilon$-dependent on a subsequence of $\{(s_1', a_1'), \ldots, (s_l', a_l')\} \subseteq \{(s_1, a_1), \ldots, (s_\kappa, a_\kappa)\}$, it implies that there exists $f, g \in \mathcal{F}^N$ with $\|f - \tilde{f}_{h,\bar{\eta}}^k\|_{\mathcal{Z}_h^k}^2 \leq \beta(\mathcal{F}^N, \delta)$ and $\|g - \tilde{f}_{h,\bar{\eta}}^k\|_{\mathcal{Z}_h^k}^2 \leq \beta(\mathcal{F}^N, \delta)$ such that $f^{(1)}(s_t', a_t') - g^{(1)}(s_t', a_t') \geq \epsilon$. By triangle inequality, $\|f - g\|_{\mathcal{Z}_h^k}^2 \leq 4\beta(\mathcal{F}^N, \delta)$. On the other hand, if $(s_j, a_j)$ is $\epsilon$-dependent on $L$ disjoint subsequences in $\{(s_1, a_1), \ldots, (s_\kappa, a_\kappa)\}$, then

$$4\beta(\mathcal{F}^N, \delta) \geq \|f - g\|_{\mathcal{Z}^k}^2 \geq \|f^{(1)} - g^{(1)}\|_{\mathcal{Z}^k}^2 \geq L\epsilon^2$$

resulting in $L \leq 4\beta(\mathcal{F}^N, \delta)/\epsilon^2$. Therefore, we have $(\kappa/\dim_E(\mathcal{F}^N, \epsilon)) - 1 \leq 4\beta(\mathcal{F}^N, \delta)/\epsilon^2$ which results in

$$\kappa \leq \left( \frac{4\beta(\mathcal{F}, \delta)}{\epsilon^2} + 1 \right) \dim_E(\mathcal{F}^N, \epsilon)$$

∎

**Lemma D.4** (Refined version of Lemma 10 in Wang et al. (2020)). *If $b_h^k(s, a) \geq w^{(1)}((\mathcal{F}^N)_h^k, s, a)$ for all $(s, a) \in \mathcal{S} \times \mathcal{A}$ and $k \in [K]$, then*

$$\sum_{k=1}^{K} \sum_{h=1}^{H} b_h^k(s_h^k, a_h^k) \leq H\dim_E(\mathcal{F}^N, 1/T).$$

*Proof.* We first sort the sequence $\{b_h^k(s_h^k, a_h^k)\}_{(k,h) \in [K] \times [H]}$ in a decreasing order and denote it by $\{e_1, \ldots, e_T\}(e_1 \geq e_2 \geq \cdots \geq e_T)$. By Lemma D.3, for any constant $M > 0$ and $e_t \geq 1/\sqrt{M}T$, we have

$$t \leq \left( \frac{4\beta(\mathcal{F}^N, \delta)}{Me_t^2} + 1 \right) \dim_E(\mathcal{F}^N, \sqrt{M}e_t) \leq \left( \frac{4\beta(\mathcal{F}^N, \delta)}{Me_t^2} + 1 \right) \dim_E(\mathcal{F}^N, 1/T)$$

which implies

$$e_t \leq \left( \frac{t}{\dim_E(\mathcal{F}^N, 1/T)} - 1 \right)^{-1/2} \sqrt{\frac{4\beta(\mathcal{F}^N, \delta)}{M}},$$

for $t \geq \dim_E(\mathcal{F}^N, 1/T)$. Since we have $e_t \leq H$,

$$\sum_{t=1}^{T} e_t = \sum_{t=1}^{T} e_t \mathbf{1}\{e_t < 1/\sqrt{M}T\} + \sum_{t=1}^{T} e_t \mathbf{1}\{e_t \geq 1/\sqrt{M}T, t < \dim_E(\mathcal{F}^N, 1/T)\}$$

$$+ \sum_{t=1}^{T} e_t \mathbf{1}\{e_t \geq 1/\sqrt{M}T, t \geq \dim_E(\mathcal{F}^N, 1/T)\}$$

$$\leq \frac{1}{\sqrt{M}} + H\dim_E(\mathcal{F}^N, 1/T) + \sum_{\dim_E(\mathcal{F}^N, 1/T) \leq t \leq T} \left( \frac{t}{\dim_E(\mathcal{F}^N, 1/T)} - 1 \right)^{-1/2} \sqrt{\frac{4\beta(\mathcal{F}^N, \delta)}{M}}$$

$$\leq \frac{1}{\sqrt{M}} + H\dim_E(\mathcal{F}^N, 1/T) + 2\left( \frac{T}{\dim_E(\mathcal{F}^N, 1/T)} - 1 \right)^{1/2} \dim_E(\mathcal{F}^N, 1/T) \sqrt{\frac{4\beta(\mathcal{F}^N, \delta)}{M}}$$

$$= \frac{1}{\sqrt{M}} + H\dim_E(\mathcal{F}^N, 1/T) + \sqrt{16 \cdot \dim_E(\mathcal{F}^N, 1/T) \cdot T \cdot \beta(\mathcal{F}^N, \delta)/M}.$$

Taking $M \to \infty$,

$$\sum_{k=1}^{K} \sum_{h=1}^{H} b_h^k(s_h^k, a_h^k) \leq H \dim_E(\mathcal{F}^N, 1/T).$$

∎

**Theorem** (6.5). *Under Assumption 4.8, with probability at least $1 - \delta$,* `SF-LSVI` *achieves a regret bound of*

$$Reg(K) \leq 2H \dim_E(\mathcal{F}^N, 1/T) + 4H\sqrt{KH \log(2/\delta)}.$$

*Proof.* Recall that $\xi_h^k = [\mathbb{P}_h(V_{h+1}^k - V_{h+1}^{\pi^k})](s_h^k, a_h^k) - (V_{h+1}^k(s_{h+1}^k) - V_{h+1}^{\pi^k}(s_{h+1}^k))$ is a martingale difference sequence where $\mathbb{E}[\xi_h^k | \mathbb{F}_h^k] = 0$ and $|\xi_h^k| \leq 2H$. By Azuma-Hoeffding's inequality, with probability at least $1 - \delta/2$,

$$\sum_{k=1}^{K} \sum_{h=1}^{H} \xi_h^k \leq 4H\sqrt{KH \log(2/\delta)}.$$

Conditioning on the above event and Lemma D.4, we have

$$\begin{aligned}
\text{Reg}(K) &\leq 2 \sum_{k=1}^{K} \sum_{h=1}^{H} b_h^k(s_h^k, a_h^k) + \sum_{k=1}^{K} \sum_{h=1}^{H} \xi_h^k \\
&\leq 2H \dim_E(\mathcal{F}^N, 1/T) + 4H\sqrt{KH \log(2/\delta)}
\end{aligned}$$

∎

