# OpenReview forum: "Bellman Unbiasedness: Toward Provably Efficient Distributional Reinforcement Learning with General Value Function Approximation"
_ICLR.cc/2025/Conference — Submitted to ICLR 2025_

### Official Review · Reviewer_kSd3 · 2024-10-29

**Soundness:** 3
**Presentation:** 2
**Contribution:** 2
**Rating:** 6
**Confidence:** 3

**Summary:**

This paper presents a regret analysis for distributional reinforcement learning (RL) with general value function approximation in finite episodic Markov decision processes (MDPs), using statistical functional dynamic programming.

Initially, it introduces the concept of Bellman unbiasedness, proving that the moment functional is the unique structure within a class that includes nonlinear statistical functionals.

The paper then addresses a new challenge related to the inherent difficulty of managing the infinite dimensionality of a distribution, offering a theoretical analysis of how hidden approximation errors hinder the development of provably efficient algorithms. The authors also revisit the distributional Bellman Completeness assumption.

Lastly, it proposes a provably efficient distributional RL algorithm named SF-LSVI, which achieves a regret bound of O(d_E H^{3/2} sqrt K) . The results are tighter and rely on weaker assumptions.

**Strengths:**

The paper is generally easy to follow.

The concept of Bellman unbiasedness is novel and interesting.

Theoretical results are achievable without assumptions of discretized rewards, small-loss bounds, or Lipschitz continuity. Compared to previous works, the regret bound is tighter, as illustrated in Table 1.

**Weaknesses:**

The purpose behind introducing the Bellman unbiasedness concept is unclear.

Assumption 4.8 appears to lack sufficient motivation. Although Bellman unbiasedness is discussed earlier in the paper, there seems to be a disconnect between it and Assumption 4.8.

It is also unclear how the model misspecification term, zeta, would enter the bounds if Assumption 4.8 does not hold. It would be worthwhile to investigate whether this would lead to a polynomial or exponential blowup.

**Questions:**

The term “Bellman closedness” generally refers to the standard RL setting (i.e., not the distributional RL setting). A brief clarification on this would be helpful.

References:

Error Bounds for Approximate Policy Iteration

Finite-Time Bounds for Fitted Value Iteration

Finite-Time Bounds for Sampling-Based Fitted Value Iteration

Information-Theoretic Considerations in Batch Reinforcement Learning

The Bellman unbiasedness concept is new to me. Does a similar concept exist in the standard RL setting? If so, is it meaningful within that context?

It seems that this paper assumes a deterministic reward. If so, can the Bellman unbiasedness concept and theoretical guarantees be extended to a stochastic reward scenario? In standard RL, extending to deterministic rewards is relatively straightforward, but I’m uncertain about how this would apply in the distributional RL setting.

In Table 1, the terms "finitely representable" and "exactly learnable" are a bit unclear. It seems these properties might limit the applicability of SF-LSVI, which could be seen as a drawback of the algorithm.

The abstract mentions "Our theoretical results demonstrate that the only way to exactly capture statistical information, including nonlinear statistical functionals, is to represent the infinite-dimensional return distribution with a finite number of moment functionals." I think it refers to Definition 4.5 and Theorem 4.6. Could you clarify its meaning?

The term "law" is unclear.

"Additional sketch" seems ambiguous.

In Definition 4.7, why is the infinity norm used? Would it be possible to use the L2 norm instead?

H' in Lemma 6.3 appears to be a typo.

What is the Dcal-norm? It is defined in the appendix but not mentioned in the main text, and it’s unclear what d represents in the appendix.

---

> ### Author Response · Authors · 2024-11-21
> **Authors' response (1/3)**
>
> Thanks for the thorough review and insightful comments on our paper. We appreciate the effort to review our work. The following is our response to the reviewer's comments.
>
>
>
> > The purpose behind introducing the Bellman unbiasedness concept is unclear. Assumption 4.8 appears to lack sufficient motivation. Although Bellman unbiasedness is discussed earlier in the paper, there seems to be a disconnect between it and Assumption 4.8.
>
> ## Motivation of Bellman unbiasedness
>
> The primary objective of our work is to explore whether it is possible to construct a provably efficient algorithm that learns not only the expected return but also the distribution of returns. When designing such algorithms, two practical constraints must be addressed:
>
> 1.	Operations must be performed in finite-dimensional spaces.
>
> 2.	Learning must rely on a finite number of sampled estimates.
>
> The early work of [Rowland et al. (2019)] introduced Bellman closedness, which ensures that operations in finite dimensions yield results consistent with the infinite-dimensional counterpart. However, as noted in their study, this property assumes knowledge of the transition kernel for the next state, which is often infeasible in practice.
>
> As mentioned in Line 80 and Section 4.2 of our paper, we define Bellman unbiasedness to address the second constraint, focusing on the practical limitations of learning from finite sampled estimates. If a sketch is not Bellman unbiased, then it implies the impossibility to eliminate the bias in the estimated sketch using a finite number of samples, resulting in consistently biased outcomes.
>
> Thus, Bellman unbiasedness is a well-motivated property that ensures unbiasedness in the results under finite-sample regimes, complementing the concept of Bellman closedness in addressing the challenges of distRL.
>
>
> ## Connection between SFBC and Bellman unbiasedness
>
> Before addressing the question in detail, we acknowledge that the sentence on Lines 344–345 is ambiguously expressed in the current manuscript. We will revise the text in the main body to eliminate any confusion. We apologize for any misunderstanding caused by this sentence, which may have implied a direct connection between Bellman unbiasedness and SFBC.
>
> **1. Clarifying the Relationship:**
>
> Bellman unbiasedness and SFBC are independent but complementary concepts for ensuring provable efficiency. To illustrate this distinction, consider the case of learning the median of the target distribution under the assumption that the transition kernel $\mathbb{P}(s'|s, a)$ is known. Since the function approximator can represent the median of the target distribution (which is just a scalar), SFBC can still be satisfied in this case. Thus, it is possible to satisfy SFBC even if the sketch is not Bellman unbiased.
> However, in the more realistic finite-sample regime where the transition kernel is unknown, the sample median of the next state does not generally match the target median. Specifically:
>
> $\mathbb{E}[\psi_{\text{median}}(\eta(s')) ] \neq \psi_{\text{median}}\big( \mathbb{E}_{s' \sim \mathbb{P}(\cdot | s, a)} \eta(s') \big)$.
>
> Moreover, as discussed in Definition 4.4 and Theorem 4.5, no transformation $\phi_\psi$ exists that can unbiasedly estimate the target median using sampled medians as input:
>
> $\nexists \phi_{\psi} \text{ such that } \mathbb{E}[ \phi_{\psi} \circ \psi_{\text{median}}(\eta(s')) ] = \psi_{\text{median}}\big( \mathbb{E}_{s' \sim \mathbb{P}(s' | s, a)} \eta(s') \big)$.
>
> This is a critical limitation of median embeddings that cannot be unbiasedly estimated through sketch-based Bellman updates, unlike mean embeddings.
>
> **2. Importance of Bellman Unbiasedness:**
>
> As highlighted in Lines 340–343, Bellman unbiasedness ensures that the sequence of sampled sketches forms a martingale. This property is essential for constructing a confidence interval around the empirical sketch, ensuring that the true target sketch lies within this interval with high probability. Such confidence intervals enable the construction of provably efficient algorithms, even under the constraints of finite samples.
>
> In summary, while Bellman unbiasedness and SFBC are independent concepts, they play complementary roles in the design of provably efficient algorithms. Bellman unbiasedness addresses the challenges of finite-sample estimation, ensuring unbiased updates, while SFBC ensures that the embedding space is expressive enough to accommodate the distributional Bellman operator.

---

> ### Author Response · Authors · 2024-11-21
> **Authors' response (2/3)**
>
> > The term “Bellman closedness” generally refers to the standard RL setting (i.e., not the distributional RL setting). A brief clarification on this would be helpful.
>
> While the term "closedness" might be interpreted differently in standard RL, the definition provided by [Rowland et al. (2019)]—*where the embedding space remains closed under the distributional Bellman operator*—is an important notion that is widely used in DistRL. In our work, we adopt this definition of Bellman closedness as it pertains specifically to finite-dimensional embeddings of infinite-dimensional return distributions.
>
> > The Bellman unbiasedness concept is new to me. Does a similar concept exist in the standard RL setting? If so, is it meaningful within that context?
>
> Bellman unbiasedness is a property designed to ensure provable efficiency in learning return distributions through various embeddings. In the case of standard RL, which deals with a single embedding—*the expected return*—this property becomes less relevant to discuss because standard RL inherently focuses on a scalar value rather than a full distribution.
>
> From the perspective of distributional RL (DistRL), the expected return can be viewed as a specific embedding (or sketch) that is already Bellman unbiased. As such, our work can be interpreted as a generalization of the standard RL setting, extending it to accommodate richer embeddings beyond the expected return.
>
> > It seems that this paper assumes a deterministic reward. If so, can the Bellman unbiasedness concept and theoretical guarantees be extended to a stochastic reward scenario? In standard RL, extending to deterministic rewards is relatively straightforward, but I’m uncertain about how this would apply in the distributional RL setting.
>
> Bellman unbiasedness is a property of the embedding (sketch) used to represent the return distribution and is therefore independent of the stochasticity of the reward. As such, this property remains valid even in extended MDPs with stochastic rewards.
> Additionally, in studies on provably efficient RL algorithms (e.g., [Jin et al. (2021)]), it is common to assume deterministic rewards for notational simplicity. This assumption does not limit the generality of the results, as the techniques can often be straightforwardly extended to stochastic reward settings, as shown in [Chen et al. (2024)].
>
> > In Table 1, the terms "finitely representable" and "exactly learnable" are a bit unclear. It seems these properties might limit the applicability of SF-LSVI, which could be seen as a drawback of the algorithm.
>
> We would like to clarify that these properties are designed to enhance, rather than limit, the applicability of SF-LSVI.
>
> **1. Finitely Representable:**
>
> As mentioned in the Introduction, infinite-dimensional distributions need to be represented with a finite number of parameters or statistical functionals, such as moments, when learning through function approximators (e.g., neural networks). Previous works often overlooked the infinite-dimensionality of distributions, assuming that a function approximator exists that can fully represent them. However, this assumption required additional constraints on the MDP to ensure feasibility.
>
> In contrast, SF-LSVI represents distributions using a finite number of sketches without requiring additional assumptions about the MDP, thereby increasing its applicability to a wider range of problems. By explicitly addressing the challenge of representing infinite-dimensional distributions, our approach removes the need for restrictive assumptions.
>
> **2. Exactly Learnable:**
>
> Even when distributions are represented in a finite-dimensional space, there is no guarantee that the corresponding Bellman operator exists for certain embeddings. For example, median and quantile embeddings do not have corresponding Bellman operators, which makes them not exactly learnable.
>
> By ensuring that the representation space aligns with a Bellman operator, SF-LSVI guarantees exact learnability. This property ensures that learning in the representation space corresponds directly to consistent updates under the distributional Bellman operator.
> In summary, these two properties are fundamental to addressing the limitations of previous approaches and contribute to the broader applicability of SF-LSVI.
>
> > The abstract mentions "Our theoretical ~ moment functionals." I think it refers to Definition 4.5 and Theorem 4.6. Could you clarify its meaning?
>
> The intended meaning is that *“Among all types of statistical functionals for representing infinite-dimensional return distributions, our theoretical results demonstrate that only moment functionals can exactly capture the statistical information.”*
> We acknowledge that the original phrasing in the abstract could be clearer, and we will revise it in future versions to avoid potential confusion.

---

> ### Author Response · Authors · 2024-11-21
> **Authors' response (3/3)**
>
> > The term "law" is unclear.
>
> In our paper, "Law" refers to the probability distribution of a random variable. Specifically, for a random variable $Z$, the notation $\text{Law}(Z)$ denotes the distribution that describes the probabilities of $Z$ taking on different values.
>
> > "Additional sketch" seems ambiguous.
>
> As illustrated in Appendix C.3 (Examples 2 and 3), variance alone cannot satisfy Bellman closedness. However, when paired with an additional sketch, such as the mean, it becomes possible to achieve Bellman closedness. In this context, the "additional sketch" refers to a complementary embedding that, when combined with the primary sketch, enables the property of Bellman closedness.
>
> On the other hand, for quantile-based embeddings, no additional sketch exists that can make them Bellman closed. This highlights a fundamental limitation of quantile embeddings in this context, as they inherently lack the structural compatibility needed for Bellman closedness, regardless of the additional information provided.
>
> > In Definition 4.7, why is the infinity norm used? Would it be possible to use the L2 norm instead?
>
> The reason we use the infinity norm is to remain consistent with the definition of misspecification error as presented in [Zanette et al. (2020)]. Their work provides a rigorous framework for analyzing how misspecification error leads to worst-case linear regret. By adopting the infinity norm, we leverage their conclusions to establish that SF-LSVI achieves linear regret under the misspecified setting. If we were to define the misspecification error using the $L_2$ norm, additional analysis would be required to characterize the impact of the error on worst-case regret.
>
> > What is the Dcal-norm? It is defined in the appendix but not mentioned in the main text, and it’s unclear what d represents in the appendix.
>
> As defined in line 735 of the appendix, $\mathcal{D}$ refers to the dataset containing state-action pairs and the corresponding $N$-dimensional data $d = [d^{(1)}, \cdots, d^{(N)} ]$. The norm $|| \cdot ||_{\mathcal{D}}$ represents the least squares loss evaluated on this dataset, capturing the error between the predicted and true values over the samples in $\mathcal{D}$.
>
> -Reference-
>
> [Jin et al. (2021)] Jin, Chi, Qinghua Liu, and Sobhan Miryoosefi. "Bellman eluder dimension: New rich classes of rl problems, and sample-efficient algorithms." Advances in neural information processing systems 34 (2021): 13406-13418.
>
> [Chen et al. (2024)]  Chen, Yu, et al. "Provable Risk-Sensitive Distributional Reinforcement Learning with General Function Approximation." arXiv preprint arXiv:2402.18159 (2024).
>
> [Zanette et al. (2020)]  Zanette, Andrea, et al. "Learning near optimal policies with low inherent bellman error." International Conference on Machine Learning. PMLR, 2020.

---

> > ### Comment · Reviewer_kSd3 · 2024-11-28
> >
> > Thanks the authors for the detailed feedback and addressing my concerns!
> >
> > I have crosschecked other reviews and I intend to maintain my score.

---

### Official Review · Reviewer_jzwz · 2024-11-03

**Soundness:** 3
**Presentation:** 2
**Contribution:** 2
**Rating:** 6
**Confidence:** 3

**Summary:**

This paper introduces the concept of Bellman unbiasedness to address limitations in distributional reinforcement learning  regarding infinite-dimensionality and model misspecification. The authors propose an algorithm, Statistical Functional Least Squares Value Iteration, that operates within a finite-dimensional space, achieving tighter regret bounds under a weaker assumption called Statistical Functional Bellman Completeness. The work leverages moment functionals as finite approximations for the distribution of returns, demonstrating that only these functionals can maintain both Bellman closedness and unbiasedness.

**Strengths:**

- Introducing Bellman unbiasedness provides a new perspective on maintaining efficiency in DistRL, specifically addressing the challenge of high-dimensional distributions.
- Theoretical rigor is high, with comprehensive proofs and well-justified assumptions.
- Key terms like Bellman closedness and unbiasedness are well-defined, and the authors provide visual aids to clarify functional relationships.

**Weaknesses:**

-  The dense theoretical sections, particularly around statistical functionals and Bellman properties, may be challenging for readers not well-versed in DistRL. Additional illustrations or simplified explanations might improve comprehension.
- The motivation of using DistRL algorithm for RL with GVFA is not strong enough. The authors provide a regret bound for the proposed algorithm, but the benifits of DistRL methods over non-DistRL methods are not shown. In addition, the comparsion to V-EST-LSR (Chen et al.,2024) may not suffice to justify the effectiveness of DistRL, given that  V-EST-LSR is designed to solve risk-sensitive tasks.
- While the paper includes theoretical analysis and basic examples, empirical validation would better demonstrate SF-LSVI’s practical utility.

**Questions:**

- How to choose $N$? How does $N$ affects the sample efficieny and computational efficiency?
- The author claims the provable efficiency of the proposed algorithm. How does the computational complexity of the proposed algorithm scale with the size of the state and action spaces, and $N$?

minor questions / comments
- Fig 1 Yellow should be Blue? why is categorical unbiased and not closed?
- Defiition 4.4, what is x_k?
- the example below Definition 4.4 seems to not involve state transition

---

> ### Author Response · Authors · 2024-11-21
> **Authors' response (1/2)**
>
> Thanks for the thorough review and insightful comments on our paper. We appreciate the effort to review our work. The following is our response to the reviewer's comments.
>
> ***
> > The dense theoretical sections, particularly around statistical functionals and Bellman properties, may be challenging for readers not well-versed in DistRL. Additional illustrations or simplified explanations might improve comprehension.
>
> We understand that concepts such as statistical functionals and Bellman properties can be challenging for readers less familiar with distributional reinforcement learning (DistRL). Figure 2 was included to provide an illustrative example of these concepts, and we are happy to elaborate on it here.
>
> **1. Sketch-Based Bellman Updates (Left Panel):**
>
> Instead of working directly with the entire distribution, the distribution is approximated using a finite-dimensional sketch $\psi$, such as the mean ($\mu$) or quantiles ($q_1, q_2, \dots$). These sketches simplify updates by representing the distribution using a fixed number of parameters.
>
> The transformation $\phi_\psi$ is applied to combine the sketches of sample distributions (e.g., $\mu, q_1, q_2, \dots$) into a sketch of the updated (mixture) distribution. This allows the update process to remain unbiased and ensures that no information is lost.
>
> **2. Ensuring Unbiasedness (Right Panel):**
>
> The right panel highlights the key property of Bellman unbiasedness. For the sketch to be unbiased, $\phi_\psi$ must accurately reconstruct the updated distribution $\psi(T\eta)$ using the sampled estimates. For example, the mean $\phi_\psi^{\text{mean}}$ successfully achieves this unbiasedness. However, as shown in Theorem 4.3, there is no transformation $\phi_\psi$ that can make quantiles $\phi_\psi^{\text{quantile}}$ unbiased. This limitation demonstrates why quantiles cannot be used in sketch-based Bellman updates under the unbiasedness framework.
>
> We appreciate your suggestion to improve accessibility. In future revisions, we plan to expand on such illustrations and include step-by-step examples to make the theoretical concepts more accessible to a broader audience.
>
> ***
> > The authors provide a regret bound for the proposed algorithm, but the benifits of DistRL methods over non-DistRL methods are not shown. In addition, the comparsion to V-EST-LSR (Chen et al.,2024) may not suffice to justify the effectiveness of DistRL, given that V-EST-LSR is designed to solve risk-sensitive tasks.
>
> Our work does not aim to explore why DistRL is inherently more effective than non-DistRL methods. Instead, our primary objective is to identify the necessary conditions for DistRL to achieve provable efficiency and to relax the structural assumptions commonly used in prior studies.
>
> For instance, while V-EST-LSR is designed to solve risk-sensitive tasks in a provably efficient manner, it relies on stricter assumptions, such as the distributional Bellman completeness (distBC) condition. To satisfy distBC, the finite-dimensional subspace must remain closed under the distributional Bellman operator, which involves translation and mixing of distributions. In practice, this requirement is challenging to achieve without strong constraints on the reward space or the MDP structure. For example, V-EST-LSR assumes discretized rewards within a linear MDP framework to satisfy these conditions.
>
> In contrast, our proposed SF-LSVI resolves these limitations by embedding the distribution into statistical functionals, such as moments, which do not require assumptions on the class of distributions or MDPs. By doing so, we provide a more general framework for achieving provable efficiency without relying on restrictive structural assumptions.
>
> The comparison to V-EST-LSR serves not to highlight the general benefits of DistRL but rather to demonstrate how SF-LSVI relaxes the structural assumptions while achieving competitive regret bounds. In this sense, we believe that V-EST-LSR is an appropriate baseline for comparison.
>
> ***
> > How to choose $N$? How does $N$ affects the sample efficieny and computational efficiency?
>
> The computational cost scales with $N$ due to the need to estimate and update multiple statistical functionals in parallel. However, this scaling is not fundamentally different from quantile-based DistRL methods, where increasing the number of quantiles similarly increases computational requirements.

---

> ### Author Response · Authors · 2024-11-21
> **Authors' response (2/2)**
>
> > The author claims the provable efficiency of the proposed algorithm. How does the computational complexity of the proposed algorithm scale with the size of the state and action spaces, and $N$?
>
> As far as we know, computational complexity is not commonly reported in GVFA research, as it can be challenging to calculate precisely due to varying solver implementations. To provide a concrete answer, we analyze the computational complexity of SF-LSVI in the specific case of linear MDPs, where such calculations are more tractable.
>
> The computational complexity primarily depends on the use of least squares regression (Algorithm 1, Line 6) and optimistic planning (Algorithm 1, Lines 7-9). In the context of linear MDPs, SF-LSVI can be viewed as a variant of LSVI algorithms, with the dimensionality of the feature map $\phi_h \colon \mathcal{S} \times \mathcal{A} \to \mathbb{R}^{d \times N}$ increasing by a factor of $N$ to accommodate multiple statistical functionals.
>
> **- Space Complexity:**
> Using the feature map $\psi_h(s, a) = \langle \phi_h(s, a), \theta_h \rangle$, the space complexity scales as $O(d  N  A  T)$, where $d$ is the dimension of the base feature space, $N$ is the number of statistical functionals.
>
> **- (Per-Step) Computational Complexity:**
> From the results in [Jin et al. (2020)], the computational complexity per timestep is $O(d^2  N^2  A  K)$, where $K$ is the number of episodes. This scaling arises due to the need to solve least squares regression for each timestep across all statistical functionals.
>
> ***
> > Fig 1 Yellow should be Blue?
>
> The (Yellow) region in Figure 1 is intended to emphasize that the conclusion from [Rowland et al. (2019)] is restricted to linear statistical functionals. However, we understand that this might lead to some confusion when interpreting the figure. To address this, we will revise the caption to represent the $(\text{Yellow})$ region as $(\text{Yellow} \cap \text{Blue})$, which better aligns with the conclusions drawn in the referenced work while maintaining clarity.
>
> ***
> > Why is categorical unbiased and not closed?
>
> According to [Rowland et al. (2019)], Lemmas 3.2 and 4.4 establish that categorical representations belong to the class of linear statistical functionals. However, they also demonstrate that categorical representations are not Bellman closed, meaning they do not satisfy the closedness property under the Bellman operator.
>
> In our work, Lemma 4.5 shows that linear statistical functionals inherently satisfy the property of being Bellman unbiased, provided that the sketch is homogeneous of degree 1. Since categorical representations meet this criterion, they are Bellman unbiased. However, their lack of closedness under the Bellman operator, as shown in [Rowland et al. (2019)], explains why categorical representations are unbiased but not closed.
>
> ***
> > Definition 4.4, what is $x_k$?
>
> The notation $x_k$ was intended as a temporary placeholder to represent the input function that takes $k$-sample sketches as its argument. Its purpose was to assist with understanding the function's input structure. However, we acknowledge that this notation might lead to unnecessary confusion. To avoid any potential misunderstandings, we will revise the text and remove this expression in future versions of the paper.
>
> ***
> > The example below Definition 4.4 seems to not involve state transition
>
> The $k$-sampled sketches are indeed results derived from state transitions, and this relationship is implicitly captured in the example. Specifically, the empirical average $\frac{1}{k} \sum_{i=1}^k$ converges to the expectation $\mathbb{E}_{\mathbb{P}}$, where the samples are generated according to the state transition.
>
> For a more precise representation, the sketches could be written as $\hat{\mu}(s'_i)$ and $\hat{\sigma}(s'_i)$, explicitly indicating their dependence on the next state $s'_i$. However, to enhance readability, we have chosen to use a more compact notation in the text. We will explicitly state that the $k$-sampled sketches are derived from state transitions.
>
> ***
> -Reference-
>
> [Jin et al. (2020)] Jin, Chi, et al. "Provably efficient reinforcement learning with linear function approximation." Conference on learning theory. PMLR, 2020.
>
> [Rowland et al. (2019)] Rowland, Mark, et al. "Statistics and samples in distributional reinforcement learning." International Conference on Machine Learning. PMLR, 2019.

---

> > ### Comment · Reviewer_jzwz · 2024-11-25
> >
> > I appreciate the authors' response, which addresses most of my concern. I have raised my score.

---

### Official Review · Reviewer_btfi · 2024-11-04

**Soundness:** 4
**Presentation:** 3
**Contribution:** 3
**Rating:** 6
**Confidence:** 3

**Summary:**

The paper introduces a novel concept in Distributional Reinforcement Learning (DistRL) called **Bellman unbiasedness**, which enables accurate and unbiased updates for statistical functionals in finite-dimensional spaces. This approach addresses the challenge of infinite-dimensional return distributions without requiring strict assumptions, such as distributional Bellman completeness. The authors propose **SF-LSVI**, an algorithm designed to achieve efficient learning with a favorable regret bound of $\tilde{O}(d_E H^{3/2} \sqrt{K})$, offering improvements over existing DistRL methods in terms of theoretical efficiency and robustness.

**Strengths:**

- **Originality**: The paper introduces Bellman unbiasedness, a novel concept that builds on Bellman closedness to allow unbiased estimation in finite-dimensional spaces. This approach opens new directions for efficient DistRL without requiring assumptions that are often infeasible in practical settings.
- **Theoretical Rigor**: The proofs and derivations are thorough and well-structured, lending strong theoretical support to the proposed SF-LSVI algorithm. The regret bound, $\tilde{O}(d_E H^{3/2} \sqrt{K})$, represents a competitive improvement within DistRL frameworks.
- **Clarity**: The paper is generally well-written, with each section logically building on the previous one. The background provided on limitations of prior approaches and the relevance of Bellman unbiasedness is helpful for contextualizing the contribution.
- **Significance**: This work could have broad implications in various applications requiring robust policy learning, such as robotics and finance, where DistRL has shown potential.

**Weaknesses:**

- **Lack of Empirical Validation**: The theoretical claims would be strengthened by empirical results on DistRL benchmarks, which could provide evidence of SF-LSVI’s practical effectiveness and robustness. Without this, the impact on real-world tasks remains speculative.
- **Completeness Assumption**: While the Statistical Functional Bellman Completeness assumption is less restrictive than full distributional completeness, it may still be challenging to meet in some environments. Further discussion on how widely this assumption holds in practical applications would be beneficial.
- **Accessibility**: Some sections, particularly the detailed theoretical derivations, may be challenging for readers not specialized in reinforcement learning. Simplifying these explanations or adding illustrative examples could make the work more accessible.

**Questions:**

1. Can the authors provide empirical validation to demonstrate SF-LSVI’s practical performance and potential advantages over existing methods?
2. How restrictive is the Statistical Functional Bellman Completeness assumption in practical settings? Could the authors discuss specific environments where this assumption may or may not hold?
3. Are there any potential limitations or specific scenarios where Bellman unbiasedness may not provide advantages or could introduce new challenges?

---

> ### Author Response · Authors · 2024-11-21
>
> Thanks for the thorough review and insightful comments on our paper. We appreciate the effort to review our work. The following is our response to the reviewer's comments.
>
> ***
> > The theoretical claims would be strengthened by empirical results on DistRL benchmarks, which could provide evidence of SF-LSVI’s practical effectiveness and robustness. Without this, the impact on real-world tasks remains speculative.
>
> As with many works in the General Value Function Approximation (GVFA) domain, our study primarily focuses on advancing theoretical foundations, such as regret analysis and the development of provably efficient algorithms, rather than empirical benchmarks. Specifically, we address the infeasibility of infinite-dimensional distributions by proposing new structural assumptions supported by rigorous theoretical guarantees.
>
> While we agree that empirical validation is important, the primary contribution of this work lies in enhancing the theoretical understanding of distributional reinforcement learning. As a natural extension, future work will involve applying SF-LSVI to established DistRL benchmarks to validate its practical utility. Additionally, we aim to extend our approach into a practical deep reinforcement learning algorithm for complex real-world environments.
>
> ***
> > How restrictive is the Statistical Functional Bellman Completeness assumption in practical settings? Could the authors discuss specific environments where this assumption may or may not hold?
>
> The SFBC assumption is an extension of the standard Bellman completeness assumption in classical RL, applied to $N$ moments of the target distribution. In practice, just as it is not considered unrealistic for a function approximator to represent the mean of the target distribution, learning $N$ moments in parallel is similarly feasible. While this does introduce additional computational cost due to the parallel estimation of multiple moments, it does not impose fundamentally restrictive requirements on the function approximator.
>
> In contrast, the Distributional Bellman Completeness (distBC) assumption is significantly more restrictive, as it requires the function approximator to represent an infinite-dimensional distribution fully. This is generally unattainable for function approximators with finite expressive power without imposing strict constraints on the MDP or the distribution class. For example, distBC assumes that all possible mixtures of distributions can be represented, which is infeasible in most practical scenarios.
>
> In summary, SFBC strikes a balance between feasibility and expressiveness by limiting the assumption to a finite number of moments, making it a more practical and achievable condition in real-world applications.
>
> ***
> > Are there any potential limitations or specific scenarios where Bellman unbiasedness may not provide advantages or could introduce new challenges?
>
> Bellman unbiasedness, along with Bellman closedness, serves as a critical property for constructing algorithms that are provably efficient and exactly learnable. However, its advantages may diminish in scenarios where the accuracy or bias of the distributional representation is not a primary concern. For instance, in cases where quantile representations are used, the mean of the estimated distribution may be biased compared to the true mean. Nonetheless, increasing the number of quantiles can effectively reduce this bias, potentially making unbiasedness less critical in such settings.
>
> On the other hand, there are specific limitations and challenges associated with the SF-LSVI framework itself. Our study assumes that the return distribution is bounded within $[0, H]$. In scenarios involving heavy-tailed distributions, the moments may not exist or may diverge, making it difficult to apply SF-LSVI effectively. In such cases, the current framework would need to be adapted or replaced with alternative approaches capable of handling distributions without well-defined moments. Exploring such extensions remains an open question for future work.

---

> > ### Comment · Reviewer_btfi · 2024-11-27
> >
> > Thank you for the response. While the theoretical contributions are significant, the lack of empirical validation limits the practical impact of SF-LSVI. Additionally, the feasibility of the SFBC assumption across diverse environments needs further exploration. I maintain my score, as the paper requires stronger practical grounding to fully justify its contributions.

---

### Official Review · Reviewer_6pLZ · 2024-11-04

**Soundness:** 3
**Presentation:** 3
**Contribution:** 3
**Rating:** 6
**Confidence:** 4

**Summary:**

This paper improves the existing distributional reinforcement learning algorithm by introducing a new concept called "Bellman unbiasedness" which is  and revisit the framework through a statistical functional lens. The paper present a new algorithm, SF-LSVI, which is provably efficient and achieves the a tight regret upper bound $\tilde{O}(d_E H^{3/2} \sqrt{K})$.

**Strengths:**

1. This paper involve a novel framework "Bellman unbiasedness" which strictly contains the existing Bellman completness assumption, and addressing the infinite-dimensionality issue in DistRL.
2. The paper proposed novel theoretical analysis towards DistRL within more general assumptions, which is a nice contribution to the RL theory community.

**Weaknesses:**

1. I think it is better to discuss more about the technical contribution,e.g., detailed introduce the technical intuition of removing the dependency $\beta$ and why the dimension term is seperate from $\sqrt{K}$.
2. It is not really clear that the relationship between the new "Bellman unbiasedness" assumption and the exist assumptions. Could you please present more comparison and examples?

**Questions:**

1. What is the notation "law" means in Lines 181 and 186.

---

> ### Author Response · Authors · 2024-11-21
> **Authors' response (1/2)**
>
> Thanks for the thorough review and insightful comments on our paper. We appreciate the effort to review our work. The following is our response to the reviewer's comments.
>
> ***
> > I think it is better to discuss more about the technical contribution,e.g., detailed introduce the technical intuition of removing the dependency $\beta$ and why the dimension term is seperate from $K$
>
> Thank you for your detailed question. We are happy to elaborate on the technical intuition behind removing the dependency on $\beta$ and why the dimension term is separate from $K$ in our final result.
>
> **1. Removing the Dependency on $\beta$:**
>
> The dependency on $\beta$ is removed through the refined decomposition outlined in Lemma D.4. Many eluder-dimension-based analyses, including those from [Wang et al. (2020)], rely on Lemma 10 from their work, which provides bounds using a bonus term dependent on $\beta$. The core improvement in our approach stems from a more precise decomposition of the bonus terms across timesteps.
>
> Specifically, we decompose the bonus terms based on whether the timestep $t$ is greater or smaller than $ \text{dim}_E(\mathcal{F}^N, 1/T) $:
>
> - When $t > \text{dim}_E(\mathcal{F}^N, 1/T)$:
> Events where the bonus term exceeds $1/\sqrt{M}T$ diminish as the constant $M$ increases.
>
> - When $t \leq \text{dim}_E(\mathcal{F}^N, 1/T)$:
> Such events remain bounded regardless of $M$.
>
> Since these results hold for any $M$, we take the infimum over $M$ to eliminate the first term, achieving a tighter bound compared to prior results.
>
> **2. Separating the Dimension Term from $K$:**
>
> We interpret the "dimension term" as referring to the Eluder dimension $\text{dim}_E(\mathcal{F}^N, 1/T)$, which characterizes the complexity of the function class $\mathcal{F}^N$ in our analysis. While it may appear that there is a dependency on $K$, since $T = KH$, we note that $\text{dim}_E(\mathcal{F}, \epsilon) = O(d \log(1/\epsilon))$ as shown in [Wang et al. (2020)]. Thus, the dependency on $K$ is only logarithmic, specifically $O(\log K)$.
>
>
> ***
> > It is not really clear that the relationship between the new "Bellman unbiasedness" assumption and the exist assumptions. Could you please present more comparison and examples?
>
> Before addressing the question in detail, we acknowledge that the sentence on Lines 344–345 is ambiguously expressed in the current manuscript. We will revise the text in the main body to eliminate any confusion. We apologize for any misunderstanding caused by this sentence, which may have implied a direct connection between Bellman unbiasedness and SFBC.
>
> **1. Clarifying the Relationship:**
>
> Bellman unbiasedness and SFBC are independent but complementary concepts for ensuring provable efficiency. To illustrate this distinction, consider the case of learning the median of the target distribution under the assumption that the transition kernel $\mathbb{P}(s'|s, a)$ is known. Since the function approximator can represent the median of the target distribution (which is just a scalar), SFBC can still be satisfied in this case. Thus, it is possible to satisfy SFBC even if the sketch is not Bellman unbiased.
> However, in the more realistic finite-sample regime where the transition kernel is unknown, the sample median of the next state does not generally match the target median. Specifically:
>
> $\mathbb{E}[\psi_{\text{median}}(\eta(s')) ] \neq \psi_{\text{median}}\big( \mathbb{E}_{s' \sim \mathbb{P}(\cdot | s, a)} \eta(s') \big)$.
>
> Moreover, as discussed in Definition 4.4 and Theorem 4.5, no transformation $\phi_\psi$ exists that can unbiasedly estimate the target median using sampled medians as input:
>
> $\nexists \phi_{\psi} \text{ such that } \mathbb{E}[ \phi_{\psi} \circ \psi_{\text{median}}(\eta(s')) ] = \psi_{\text{median}}\big( \mathbb{E}_{s' \sim \mathbb{P}(s' | s, a)} \eta(s') \big)$.
>
> This is a critical limitation of median embeddings that cannot be unbiasedly estimated through sketch-based Bellman updates, unlike mean embeddings.
>
> **2. Importance of Bellman Unbiasedness:**
>
> As highlighted in Lines 340–343, Bellman unbiasedness ensures that the sequence of sampled sketches forms a martingale. This property is essential for constructing a confidence interval around the empirical sketch, ensuring that the true target sketch lies within this interval with high probability. Such confidence intervals enable the construction of provably efficient algorithms, even under the constraints of finite samples.
>
> In summary, while Bellman unbiasedness and SFBC are independent concepts, they play complementary roles in the design of provably efficient algorithms. Bellman unbiasedness addresses the challenges of finite-sample estimation, ensuring unbiased updates, while SFBC ensures that the embedding space is expressive enough to accommodate the distributional Bellman operator.

---

> ### Author Response · Authors · 2024-11-21
> **Authors' response (2/2)**
>
> > What is the notation "law" means in Lines 181 and 186.
>
> In our paper, "Law" refers to the probability distribution of a random variable. Specifically, for a random variable $Z$, the notation $\text{Law}(Z)$ denotes the distribution that describes the probabilities of $Z$ taking on different values.
>
> ***
> -Reference-
>
> [Wang et al. (2020)] Wang, Ruosong, Russ R. Salakhutdinov, and Lin Yang. "Reinforcement learning with general value function approximation: Provably efficient approach via bounded eluder dimension." Advances in Neural Information Processing Systems 33 (2020): 6123-6135.

---

> > ### Comment · Reviewer_6pLZ · 2024-11-27
> >
> > Thanks for your detailed response. Most of my concerns have been addressed. I choose to maintain my score and recommend acceptance.

---

### Official Review · Reviewer_QFTP · 2024-11-05

**Soundness:** 4
**Presentation:** 3
**Contribution:** 2
**Rating:** 3
**Confidence:** 3

**Summary:**

This paper designs a distributional reinforcement learning algorithm with general function approximation, called SF-LSVI, under the assumption that the Eluder dimension of the function class is small. In particular, the SF-LSVI algorithm estimates the momentums of the expected return in addition to its mean (which is the standard Q-function). This paper proves that the SF-LSVI algorithm achieves a near-optimal regret bound.

**Strengths:**

-	As one of the major technical contributions, this paper studies the distributional RL problem with general reward functions, whereas prior works mostly focus on discretized rewards. This paper also identifies two key assumptions, Bellman closedness and Bellman unbiasedness, that allow statistically efficient learning algorithms.
-	The choice of the particular choice of the sketch function, namely, the momentums, is theoretically justified by Theorem 4.6, which proves that “the only finite statistical functionals that are both Bellman unbiased and closed … is equal to the linear span of the set of moment functionals”.

**Weaknesses:**

-	While this paper studies distributional RL, the final metric for the performance of the algorithm is still the standard regret. In the main theorem (Theorem 6.5), there is no guarantee of whether the estimated momentum is close to the ground truth. In fact, estimating the momentums is purely an independent component in the algorithm, and is independent of learning the optimal policy.

**Questions:**

-	The justification for Bellman unbiasedness is confusing to me. Why is Bellman unbiasedness necessary in the finite sample regime? Is it a fundamental requirement (in other words, there are some statistical lower bounds if Bellman unbiasedness does not hold) or just a technical assumption required by the current analysis?

---

> ### Author Response · Authors · 2024-11-21
>
> Thanks for the thorough review and insightful comments on our paper. We appreciate the effort to review our work. The following is our response to the reviewer's comments.
>
> ***
> >  While this paper studies distributional RL, the final metric for the performance of the algorithm is still the standard regret. In the main theorem (Theorem 6.5), there is no guarantee of whether the estimated momentum is close to the ground truth.
>
> Our work focuses on developing a provably efficient algorithm for learning distributional information rather than directly addressing why distributional RL (DistRL) may be more advantageous compared to standard RL. While regret is widely employed as a measure of provable efficiency, it does not directly indicate how well the distributional moments match the ground truth.
>
> That said, our proposed algorithm, SF-LSVI, performs moment least squares regression, as detailed in Lemmas 6.3 and 6.4. These lemmas show that all $N$ moments lie within the confidence intervals (as defined in the norm described in Table 2 of Appendix A, which aggregates the $N$-moment estimates). Thus, while regret does not explicitly measure the accuracy of moment estimation, SF-LSVI inherently guarantees that all moments converge closer to the ground truth.
>
> To address the limitations of standard regret, we proposed a future direction in Line 522: *redefining regret to capture not just the expected value but also all moments.* This extension could provide a more comprehensive evaluation of distributional algorithms. Nonetheless, for the sake of fair comparisons with existing methods, we adhered to the standard regret metric in this work.
> We hope this clarifies our approach and highlights the importance of moment estimation guarantees provided by SF-LSVI.
>
> ***
> >  The justification for Bellman unbiasedness is confusing to me. Why is Bellman unbiasedness necessary in the finite sample regime?
>
> The primary objective of our work is to explore whether it is possible to construct a provably efficient algorithm that learns not only the expected return but also the distribution of returns. When designing such algorithms, two practical constraints must be addressed:
>
> 1.	Operations must be performed in finite-dimensional spaces.
>
> 2.	Learning must rely on a finite number of sampled estimates.
>
> The early work of [Rowland et al. (2019)] introduced Bellman closedness, which ensures that operations in finite dimensions yield results consistent with the infinite-dimensional counterpart. However, as noted in their study, this property assumes knowledge of the transition kernel for the next state, which is often infeasible in practice.
>
> As mentioned in Line 80 and Section 4.2 of our paper, we define Bellman unbiasedness to address the second constraint, focusing on the practical limitations of learning from finite sampled estimates. If a sketch is not Bellman unbiased, then it implies the impossibility to eliminate the bias in the estimated sketch using a finite number of samples, resulting in consistently biased outcomes.
>
> For example, in finite-sample regime where the transition kernel is unknown, the sample median of the next state does not generally match the target median. Specifically:
>
> $\mathbb{E}[\psi_{\text{median}}(\eta(s')) ] \neq \psi_{\text{median}}\big( \mathbb{E}_{s' \sim \mathbb{P}(\cdot | s, a)} \eta(s') \big)$.
>
> Moreover, as discussed in Definition 4.4 and Theorem 4.5, no transformation $\phi_\psi$ exists that can unbiasedly estimate the target median using sampled medians as input:
>
> $\nexists \phi_{\psi} \text{ such that } \mathbb{E}[ \phi_{\psi} \circ \psi_{\text{median}}(\eta(s')) ] = \psi_{\text{median}}\big( \mathbb{E}_{s' \sim \mathbb{P}(s' | s, a)} \eta(s') \big)$.
>
> This is a critical limitation of median embeddings that cannot be unbiasedly estimated through sketch-based Bellman updates, unlike mean embeddings.
>
> Furthermore, Bellman unbiasedness ensures that the sequence of sampled sketches forms a martingale. This property is essential for constructing a confidence interval around the empirical sketch, ensuring that the true target sketch lies within this interval with high probability. Such confidence intervals enable the construction of provably efficient algorithms, even under the constraints of finite samples.
>
> Thus, Bellman unbiasedness is a well-motivated property that ensures unbiasedness in the results under finite-sample regimes, complementing the concept of Bellman closedness in addressing the challenges of distRL.
>
> ***
> -Reference-
>
> [Rowland et al. (2019)] Rowland, Mark, et al. "Statistics and samples in distributional reinforcement learning." International Conference on Machine Learning. PMLR, 2019.

---

> > ### Comment · Reviewer_QFTP · 2024-11-23
> >
> > Thank you for the response. While I do believe that distributional RL algorithms in general have great potential, I still think that decoupling the distributional aspect and the decision-making aspect over-simplifies the problem, which weakens the theoretical question studied in this paper. If the goal is to minimize regret and estimate the momentums of the cumulative reward generated by the policy, one can simply run a standard RL algorithm to find a near-optimal policy, and then estimate the momentums of the return by rolling out the policy multiple times.
> >
> > Therefore, I still believe that using the standard regret as the metric for the algorithm is a significant weakness of this paper.

---

> > > ### Author Response · Authors · 2024-11-23
> > >
> > > Thank you for your thoughtful feedback.
> > >
> > > Regret analysis is specifically designed for online learning scenarios, where the goal is to iteratively improve the policy while minimizing cumulative regret. Performing additional rollouts to gather auxiliary information about a near-optimal policy, as suggested, is impractical in these settings, as the learning process must occur dynamically and in real time.
> > >
> > > For instance, in robotics control, a robot must adapt its actions based on immediate feedback from the environment. Interrupting this process to perform additional rollouts would disrupt real-time learning and is often infeasible due to environmental dynamics and time constraints. Similarly, in online advertising, decisions must be made in real time based on streaming user data to maximize click-through rates (CTR). Rollouts in this context would delay decisions and result in missed opportunities for optimization.
> > >
> > > **Purpose of the Paper:**
> > > Our primary objective is **to establish feasible conditions** under which provable efficiency can be achieved in distRL. This work does not aim to directly address why distRL may be more beneficial than standard RL; instead, it lays the theoretical groundwork necessary for designing provably efficient algorithms under finite-sample constraints.
> > >
> > > **Future Directions:**
> > > We agree that redefining regret to better capture the practical benefit of distRL is an important direction for future research.
> > > However, any such reformulated framework must build upon the conditions established in our work. Without satisfying properties such as Bellman unbiasedness or Statistical Functional Bellman Completeness, achieving provable efficiency in DistRL would not be feasible. For instance, in risk-sensitive RL, even when using a distributional approach, moment embeddings remain essential to avoid biased results. This highlights the foundational importance of the theoretical principles introduced in our paper for ensuring the practicality and reliability of DistRL methods.
> > >
> > > We hope this response clarifies the scope and contributions of our work, as well as its relevance to future research directions.

---

> > > > ### Comment · Reviewer_QFTP · 2024-11-27
> > > >
> > > > Thank you for your additional response. I am still not convinced that the theoretical result of this paper is significant enough, and I will elaborate on my points in the following.
> > > >
> > > > > Performing additional rollouts to gather auxiliary information about a near-optimal policy, as suggested, is impractical in these settings, as the learning process must occur dynamically and in real time.
> > > >
> > > > I generally don’t think performing rollouts is a problem both theoretically and empirically. It is well-known that low regret can be achieved by low switching algorithms (e.g., [1]). In other words, frequent policy updates are not necessary, and rollouts are just some steps of the RL algorithm without policy updates.
> > > >
> > > > > Our primary objective is to establish feasible conditions under which provable efficiency can be achieved in distRL.
> > > >
> > > > My major concern is that this paper lacks a clear theoretical definition of what is considered “achieving distRL”. The current definition (low regret + estimating the momentums) can be achieved by a simple alternative algorithm that does not use any of the given conditions.
> > > >
> > > > In addition, I am also not convinced that the analysis in this paper reflects the principle of designing empirical RL algorithms. Finding the confidence region for neural networks is generally a very challenging task. Therefore, it is not a popular design choice for empirical algorithms.
> > > >
> > > > Therefore, I will keep my score for now. If the authors can find a clear theoretical separation between the proposed algorithm SF-LSVI and other straightforward alternative algorithms, I am willing to re-evaluate this paper.
> > > >
> > > > [1] Bai, Yu, et al. Provably efficient q-learning with low switching cost.

---

> ### Author Response · Authors · 2024-11-28
>
> Thank you for your thoughtful feedback.
> Your main concern appears to be the theoretical distinction between SF-LSVI and alternative algorithms, such as standard RL methods that separately estimate sketches like moments.
> We would like to address this in detail:
>
> **Theoretical Foundations of Sketch-based Bellman Updates**
>
> A key contribution lies in *uncovering the fundamental limitations of embeddings when estimating Bellman targets with finite samples.*
> As demonstrated in Theorem 4.6, the only feasible embedding for unbiasedly estimating Bellman targets is the moment embedding. Similarly, Theorem 4.3 shows that quantile embeddings lack a corresponding Bellman update and thus cannot be used to estimate Bellman targets.
>
> This result applies universally to any sketch-based algorithm, including the alternative approach you mentioned.
> Even in algorithms that separately estimate moments, moments remain the only valid embeddings for unbiased Bellman updates.
> SF-LSVI builds on this foundational insight, designing an algorithm specifically tailored to these constraints, and serves as a prototype for provably efficient distributional RL under finite-sample settings.
>
> **Convergence of Moment Estimation**
>
> SF-LSVI further distinguishes itself through its *theoretical guarantees on the convergence of moment estimates.*
> As established in Theorem 6.3, SF-LSVI ensures that the sum of the confidence intervals for moments converges at a rate of $O(\sqrt{T})$, even when policies are updated.
> This means that the moment estimates *maintain* theoretical guarantees of optimality and unbiasedness, converging to the true sketch of the Bellman target.
>
> In contrast, alternative algorithms based on standard RL methods with separate moment estimation steps lack such guarantees. After each policy update, these approaches must *re-estimate* moments for the new distribution through rollouts, as previously learned moment information is not retained. This necessitates additional analyses to evaluate the convergence rates of confidence intervals for these moments, which can depend heavily on rollout length and approximation quality.
>
> This is precisely where SF-LSVI's *moment least-squares regression* distinguishes itself.
> By integrating moment estimation directly into the learning process, SF-LSVI ensures that previously gathered information is retained and effectively utilized during subsequent policy updates. This approach not only eliminates the need for additional rollouts but also provides guarantees for the optimal convergence rates of confidence intervals, as established in Theorem 6.3.
>
> For example, if the rollout length is bounded by a constant $c$, each confidence interval will incur an approximation error of $\epsilon(c)$. Consequently, the cumulative confidence interval grows as $\epsilon(c) \cdot T $, which does not achieve the $O(\sqrt{T})$ convergence rate of SF-LSVI. This trade-off between rollout length and moment estimation accuracy complicates the design of alternative algorithms and further highlights the efficiency of SF-LSVI.
>
> **Finding the confidence region for neural networks is generally a very challenging task**
>
> It is true that finding an exact confidence interval width can be computationally challenging, as it often involves solving NP-hard set-constrained optimization problems. However, as noted in Appendix B: Technical Remarks, many GVFA studies have successfully proposed confidence-based algorithms despite this complexity. Notably, the width of the confidence interval primarily serves to encourage optimistic exploration, rather than directly determining regret performance.
> As shown by [Abbasi-Yadkori et al. (2011)], approximation errors are known to have a small effect on regret. This suggests that even approximate confidence regions can provide sufficient guidance for efficient exploration, supporting the feasibility of confidence-based methods in practice.
>
>
>
>
> **Conclusion**
>
> In summary, SF-LSVI is uniquely designed to leverage the special properties of moment embeddings for unbiased and efficient Bellman updates. It provides guarantees on both the feasibility of sketch-based updates and the convergence of moment estimates, which are absent in alternative algorithms. These theoretical distinctions demonstrate the significance of SF-LSVI in advancing the foundation of provably efficient distRL.
>
> We hope this clarifies our position and demonstrates that confidence regions, while challenging to compute exactly, remain a practical and theoretically grounded tool for distributional RL. Please let us know if additional clarification is needed.
>
> -References-
>
> [Abbasi-Yadkori et al. (2011)] Abbasi-Yadkori, Yasin, Dávid Pál, and Csaba Szepesvári. "Improved algorithms for linear stochastic bandits." Advances in neural information processing systems 24 (2011).

---

### Meta-Review · Area_Chair_Fnbe · 2024-12-22

**Metareview:**

This paper studies distributional reinforcement learning with general value function approximation. The authors design a distributional RL algorithm with general function approximation, called SF-LSVI, and prove that SF-LSVI  achieves a near-optimal regret bound under the assumption that the Eluder dimension of the function class is small.

The main weakness raised by the reviewers is the theoretical formulation of the distributional RL problem. In this paper, the goal is to achieve a small regret, which is independent of the distributional RL framework. Therefore, the problem formulation might not reflect the real use case of distributional RL.

Given the high standards of ICLR and the weakness mentioned above, I would recommend rejecting this paper.

**Additional Comments On Reviewer Discussion:**

The reviewers raised concerns regarding the theoretical formulation of the distributional RL problem, the lack of motivation for the Bellman unbiasedness concept, as well as comparison between Bellman unbiasedness and existing assumptions. Although the authors provided responses which addressed some of those concerns, concerns regarding the theoretical formulation of the distributional RL problem remain.

---

### Decision · Program_Chairs · 2025-01-22

Reject